# Neurofibromin 1 controls metabolic balance and Notch-dependent quiescence of murine juvenile myogenic progenitors

Xiaoyan Wei[1,2], Angelos Rigopoulos [1,2,3], Matthias Lienhard [4], Sophie Pöhle-Kronawitter [1], Georgios Kotsaris [1], Julia Franke[1,2], Nikolaus Berndt[5,6,7], Joy Orezimena Mejedo [1], Hao Wu [8], Stefan Börno[9], Bernd Timmermann[9], Arunima Murgai [1], Rainer Glauben[8] & Sigmar Stricker [1,2,3] ✉

Patients affected by neurofibromatosis type 1 (NF1) frequently show muscle weakness with unknown etiology. Here we show that, in mice, Neurofibromin 1 (*Nf1*) is not required in muscle fibers, but specifically in early postnatal myogenic progenitors (MPs), where *Nf1* loss led to cell cycle exit and differentiation blockade, depleting the MP pool resulting in reduced myonuclear accretion as well as reduced muscle stem cell numbers. This was caused by precocious induction of stem cell quiescence coupled to metabolic reprogramming of MPs impinging on glycolytic shutdown, which was conserved in muscle fibers. We show that a Mek/Erk/NOS pathway hypersensitizes *Nf1*-deficient MPs to Notch signaling, consequently, early postnatal Notch pathway inhibition ameliorated premature quiescence, metabolic reprogramming and muscle growth. This reveals an unexpected role of Ras/Mek/Erk signaling supporting postnatal MP quiescence in concert with Notch signaling, which is controlled by Nf1 safeguarding coordinated muscle growth and muscle stem cell pool establishment. Furthermore, our data suggest transmission of metabolic reprogramming across cellular differentiation, affecting fiber metabolism and function in NF1.

During prenatal development, myogenic progenitors (MPs) that originate from the dermomyotome compartment of the somitic mesoderm migrate to their terminal destinations in the trunk and limbs. Here, MPs proliferate to expand the progenitor pool, from where cells continuously exit the cell cycle and differentiate into myoblasts under the control of myogenic regulatory factors Myf5, Mrf4 (Myf6), MyoD, and myogenin (Myog). Myoblasts ultimately fuse into multinucleated myofibers[1,2]. After birth, myofibers continue to grow by myonuclear accrual, carried by MPs that express the transcription factor Pax7. This process, however, gradually reduces and is replaced by metabolic fiber

[1]Musculoskeletal Development and Regeneration Group, Institute of Chemistry and Biochemistry, Freie Universität Berlin, 14195 Berlin, Germany. [2]Max Planck Institute for Molecular Genetics, 14195 Berlin, Germany. [3]International Max Planck Research School for Biology and Computation IMPRS-BAC, Berlin, Germany. [4]Department of Computational Molecular Biology, Max Planck Institute for Molecular Genetics, 14195 Berlin, Germany. [5]Department of Molecular Toxicology, German Institute of Human Nutrition Potsdam-Rehbruecke (DIfE), Nuthetal, Germany. [6]Institute of Computer-assisted Cardiovascular Medicine, Deutsches Herzzentrum der Charité (DHZC), Berlin, Germany. [7]Charité – Universitätsmedizin Berlin, corporate member of Freie Universität Berlin and Humboldt-Universität zu Berlin, Berlin, Germany. [8]Division of Gastroenterology, Infectiology and Rheumatology, Medical Department, Charité University Medicine Berlin, 12203 Berlin, Germany. [9]Sequencing Core Unit, Max Planck Institute for Molecular Genetics, 14195 Berlin, Germany. ✉e-mail: sigmar.stricker@fu-berlin.de

growth[3]. Along with reduced myonuclear accrual, Pax7[+] MPs exit the cell cycle and become quiescent. In parallel, they assume their characteristic position beneath the myofiber basal lamina; hence, they were called "satellite cells"[4,5]. Adult satellite cells are muscle stem cells (MuSCs) that remain quiescent until activation by, for example, acute injury, after which they re-enter the cell cycle to establish a new MP pool to regenerate myofibers[6].

In mice, most Pax7[+] juvenile MPs enter quiescence by postnatal day 21 (p21)[3,7]. However, few proliferative Pax7[+] MPs are still present until approx. p56[8] and even after, contributing to low steady-state muscle turnover[9,10]. Quiescence of adult MuSC is maintained by a combination of niche signals, disruption of which leads to MuSC activation and cell cycle entry[11–18]. The primal establishment of MuSC quiescence during early postnatal life, however, is less understood. Notch signaling has been involved, as genetic perturbation disrupts MPs' homing to the MuSC niche[19] and disturbs the cell cycle exit of juvenile MPs and their transition to quiescence[20–22].

*Neurofibromin 1* (*Nf1*) is a tumor suppressor gene that encodes a RAS–guanosine triphosphatase–activating protein (RAS-GAP) that reduces RAS activity and inhibits mitogen-activated protein kinase 1/2–extracellular signal-regulated protein kinase 1/2 (Mek1/2-Erk1/2) downstream signaling[23]. Autosomal-dominant *NF1* mutations cause neurofibromatosis type 1 (NF1), one of the most common genetic diseases affecting approx. 1 in 3000–4000 live births, which is mainly characterized by the presence of nerve sheath tumors[23], in addition to musculoskeletal involvement, strongly affecting the patients' mobility and quality of life[24]. NF1 patients, as well as those with mutations in other members of the RAS pathway (so-called RASopathies), often display severe muscle weakness[25–27]. The function of *Nf1* in muscle is unclear. Constitutive inactivation of *Nf1* causes early embryonic lethality[28], while *Nf1* haploinsufficiency does not affect muscle development or function in mice[29]. Widespread *Nf1* inactivation in limb mesenchyme reduces muscle size and function[30]. Muscle-specific *Nf1* inactivation via Myod[Cre] causes early postnatal lethality[29], while *Nf1* inactivation via Myf5[Cre] causes viable offspring with muscle hypotrophy, fast fiber atrophy, and a whole-body catabolic phenotype[31]. Intriguingly, all three models indicated aberrant muscle metabolism with distorted carbohydrate and lipid usage[29,31,32].

In this study we show that *Nf1* is required in murine juvenile MPs to prevent metabolic reprogramming and precocious induction of stem cell quiescence. Loss-of *Nf1* amplifies a Ras-Mek1/2-Erk1/2 signaling axis that funnels into Notch signaling. This distorts the balance between MP amplification/differentiation vs. quiescence induction, draining the postnatal MP pool, thus affecting postnatal muscle growth and MuSC pool establishment. Importantly, Nf1 was dispensable in muscle fibers, indicating that metabolic reprogramming of MPs can be transmitted to adult myofibers, and that the NF1-associated myopathy is a postnatal developmental disease.

## Results

### Premature cell cycle exit and impaired myogenic differentiation of Nf1[Myf5] MPs reduces myonuclear accrual and MuSC numbers

Mice with conditional inactivation of *Nf1* targeted to myoblasts by using Myf5[Cre] (Myf5[Cre];Nf1[flox/flox], or "Nf1[Myf5]") reduce mTORC1-dependent anabolic myofiber growth during postnatal development in a variety of fore- and hind limb muscles, already visible during the first 3 weeks of postnatal life[31]. During juvenile development, however, muscle growth occurs by a combination of myonuclear cell accrual and metabolic growth[3,7,8], so we analyzed Pax7[+] juvenile MP behavior in Nf1[Myf5] mice. Reverse transcription–quantitative polymerase chain reaction (RT-qPCR) and RNA-sequencing (RNA-Seq) confirmed efficient decrease in *Nf1* messenger RNA (mRNA) in fluorescence-activated cell sorting (FACS)-isolated MPs at postnatal day 7 (p7) (Supplementary Fig. 1a). Nf1 expression was unaltered in p14 MPs of *Nf1*-haploinsufficient Myf5[Cre];Nf1[flox/+] mice (Supplementary Fig. 1b),

which were used as controls for all further experiments; male and female animals were mixed in both groups. Myf5[Cre] mice are haploinsufficient for *Myf5*; however, *Myf5* expression was not affected in p7 MPs of Myf5[Cre/+] mice (Supplementary Fig. 1c). Both observations overlap our previous findings from Nf1[Myf5] muscle tissue[31], suggesting that common compensatory mechanisms are in place.

Proportions of proliferating of Pax7[+] cells assessed by Ki67 immunolabeling of tissue sections showed a slight reduction in Nf1[Myf5] muscle at p7, with a high decrease at p14 (Fig. 1a). At p21, Pax7[+] cells appeared mostly non-proliferative in Nf1[Myf5] muscles (Fig. 1a, b). A low fraction of adult MuSCs is in the cell cycle[33], which was also seen at p84 in controls but not in Nf1[Myf5] Pax7[+] MuSCs (Fig. 1a). Freshly FACS-isolated p14 MPs after cytospin confirmed a reduced fraction of proliferating cells (Fig. 1c). In addition, p14 Nf1[Myf5] MPs showed increased Pax7 protein abundance, indicated by fluorescence intensity measurement (Fig. 1d).

Cytospun p14 MPs showed a relative decrease in MyoD[+]/Pax7[+] cell numbers (Fig. 1e). Freshly isolated Nf1[Myf5] p14 MPs plated in high density and immediately subjected to differentiation conditions showed a strong decrease in myotube formation compared to control MPs (Fig. 1f). No alterations in proliferation rate or Pax7/MyoD ratio was observed in haploinsufficient Myf5[Cre];Nf1[flox/+] p14 MPs (Supplementary Fig. 1d). In summary, Nf1Myf5 MPs were less proliferative than controls, and following activation in vitro showed impaired differentiation.

Decreased proliferation and differentiation led to reduced myonuclear accrual in Nf1[Myf5] mice, as shown by reduced myonuclear numbers in single fast (MyHC-2B[+]) fibers from extensor digitorum longus (EDL) muscles of adult (15-week-old) mice (Fig. 1g). In addition, we found a reduced myonuclear domain (the amount of cytoplasm allocated to one myonucleus) in Nf1[Myf5] fibers (Fig. 1g), suggesting no compensatory myonuclear domain growth[34] occurred in this model, consistent with the metabolic growth defect of Nf1[Myf5] muscle[31]. Pax7[+] cell numbers of Nf1[Myf5] muscles were normal at p7 but reduced in the following 2 weeks of postnatal life, and at p21, Pax7[+] cell numbers in Nf1[Myf5] muscles reduced to ~50% of control levels (Fig. 1h) and remained constant thereafter, as found at p84 (Fig. 1h), indicating a lasting decrease in MuSC numbers. We did not detect aberrant apoptosis in Nf1[Myf5] muscle (Supplementary Fig. 1e).

We conclude that precocious postnatal cell cycle withdrawal of Pax7[+] MPs and a differentiation blockade explain the decrease in MP numbers and myonuclear accretion, as well as the diminished adult MuSC pool in *Nf1[Myf5]* mutants.

### Nf1 is dispensable in muscle fibers

Myf5[Cre] targets myogenic progenitors (myoblasts), thus leading to an early recombination in the majority of the myogenic lineage[35]. We first analyzed *Nf1* expression during myogenic differentiation. *Nf1* gene expression decreased during myogenic differentiation of primary mouse myoblasts (Fig. 2a). In addition, in freshly isolated p7 MPs, *Nf1* mRNA was less abundant compared to p21 MPs (Fig. 2b). *Nf1* mRNA was present in p21 whole muscle tissue (Fig. 2b); however, adherent fibroblastic populations appeared as the major source of this expression (Fig. 2b). To disentangle the function of *Nf1* within myofibers uncoupled from an earlier function in MPs, we inactivated *Nf1* using Acta1[Cre], which targets myofibers but not myoblasts[36] via expression of Cre from a transgene driven by the human skeletal actin promoter. Acta1[Cre] specificity in limb muscle fibers, but not Pax7[+] myogenic progenitors (MPs), was confirmed in Rosa26[mTmG] reporter mice[37] (Supplementary Fig. 2a, b). RT-qPCR confirmed the efficiency of *Nf1* deletion in p21 muscle tissue (Supplementary Fig. 2c). Surprisingly, Acta1[Cre];Nf1[flox/flox] mice (Nf1[Acta1]) showed normal growth and were indistinguishable from littermates (Fig. 2c). The whole muscle cross-sectional area and fiber diameters of TA, EDL and Triceps muscles of Nf1[Acta1] mice were equal to controls (Fig. 2d, e, Supplementary Fig. 2d). While Nf1[Myf5] muscles showed a fiber type shift and altered metabolic

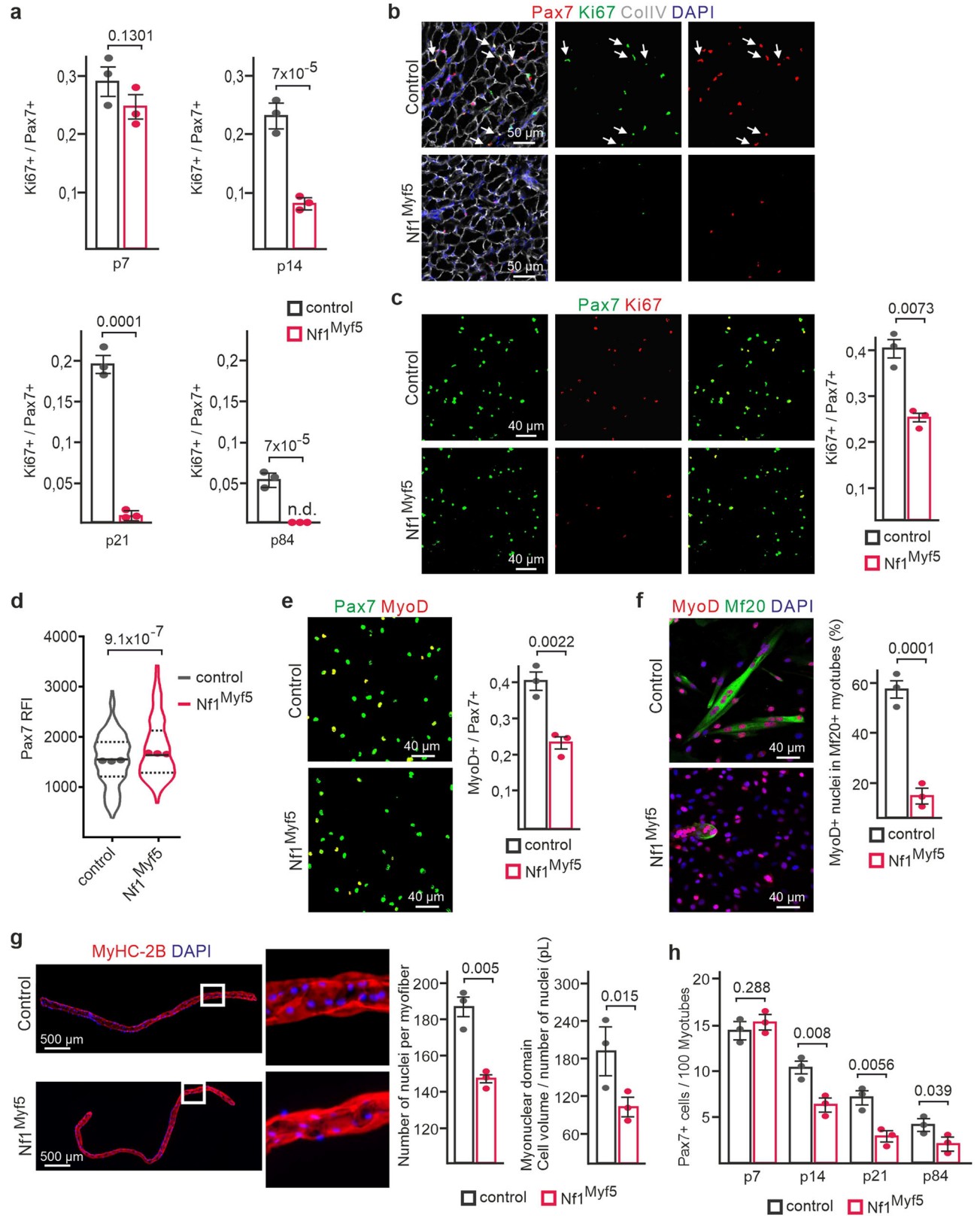

gene expression profiles[31], no changes in relative numbers of Type 1, Type 2A or Type 2B fibers were found in Nf1[Acta1] TA or EDL muscles (Fig. 2f, g), and no alteration in a selection of metabolic genes that were deregulated in Nf1[Myf5] muscle[31] was found (Supplementary Fig. 2e).

We conclude that Nf1 is downregulated during myogenic differentiation and is not required in mature muscle fibers, suggesting that myopathy of Nf1[Myf5] animals was caused by aberrant progenitor programming.

### Nf1-deficient MPs are shifted to premature quiescence

Cell cycle exit with a lack of differentiation and increased Pax7 expression indicates a shift of MPs to a quiescent phenotype. To further address this at the phenotype onset, we analyzed freshly FACS-

**Fig. 1 | Premature cell cycle exit and impaired differentiation of Nf1^Myf5 MPs reduces myonuclear accrual and MuSC numbers. a** Ki67^+/Pax7^+ cells quantification relative to all Pax7^+ cells in TA muscles of control and Nf1^Myf5 mice at indicated time points. p, postnatal day (n = 3 animals per genotype; p-values shown). **b** Representative immunolabeling images of Pax7 (red), Ki67 (green), and DAPI (nuclei, blue) of p21 muscle sections of control or Nf1^Myf5 mice. Arrows indicate Pax7^+/Ki67^+ cells (n = 3 animals per genotype). **c** Cytospin of FACS-isolated p14 MPs from control or Nf1^Myf5 mice labeled for Pax7 (green), Ki67 (red), and DAPI (nuclei, blue). Quantification of Ki67^+/Pax7^+ cells relative to all Pax7^+ cells shown right (n = 3 animals per genotype; p-value shown). **d** Quantification of anti-Pax7 relative fluorescence intensity (RFI) on images as in (**c**). Data range is shown as violin plot with median and interquartile range, means of biological replicates are shown as dots (n = 3 animals per genotype; p-value shown). **e** Cytospin of FACS-isolated p14 MPs from control or Nf1^Myf5 mice labeled for Pax7 (green), MyoD (red), and DAPI (nuclei, blue). Quantification of MyoD^+/Pax7^+ cells (right) (n = 3 animals per genotype; p-value shown). **f** In vitro differentiation of FACS-isolated p14 MPs from control or Nf1^Myf5 mice after 2 d differentiation stained for Myosin (Mf20, green), MyoD (red) and DAPI (nuclei, blue). Quantification of MyoD^+ nuclei within Mf20+ myotubes relative to all MyoD^+ nuclei (right) (n = 3 animals per genotype; p-value shown). **g** Left: Representative images of single fibers isolated from 15-week EDL muscles stained for MyHC-2B (red) and DAPI (nuclei, blue). Boxed region shown as magnification. Right: Quantification of nuclei per myofiber and myonuclear domain (cell volume/number of nuclei); pL picoliter, (n = 3 animals per genotype; p-values shown). **h** Pax7^+ cell quantification on sections of TA muscles of control or Nf1^Myf5 mice at indicated time points (n = 3 animals per genotype; p-values shown). Data are mean ± SEM; P-value calculated by two-sided unpaired t-test. Source data are provided as a Source Data file.

isolated Nf1^Myf5 and control p7 MPs by RNA-Seq. Two biological replicates, each consisting of cells pooled from 2 mice, were used for each genotype (Supplementary Fig. 3a, b, Supplementary Data 1). Gene set enrichment analysis (GSEA) showed an enrichment for NRAS Signaling in Nf1^Myf5 MPs (Fig. 3a) in line with upregulated RAS pathway activity, and p-Erk immunolabeling intensity was increased in cytospun Nf1^Myf5 MPs (Fig. 3b). Consistent with the reduced differentiation potential of Nf1^Myf5 MPs, myogenesis-related GSEA terms were enriched in controls (Fig. 3c, Supplementary Fig. 3c). In contrast, GSEA terms associated with the ECM and basal lamina, both essential for MuSC quiescence[11,38], were overrepresented in mutants (Supplementary Fig. 3d).

DESeq2 analysis confirmed downregulation of myogenic differentiation genes and upregulation of Pax7 and other quiescence-related genes in Nf1^Myf5 MPs (Fig. 3d). Kyoto Encyclopedia of Genes and Genomes (KEGG) analysis of differentially regulated genes confirmed "ECM–receptor interaction" and "Focal adhesion" among the highest-enriched terms in genes upregulated in Nf1^Myf5 MPs, while genes downregulated in Nf1^Myf5 MPs showed enrichment for numerous terms related to cellular metabolism (Supplementary Fig. 3e).

We compiled MuSC activation and quiescence signatures, for which published transcriptome datasets[11,39–43] were mined for genes commonly down- or upregulated. This yielded 142 activation-associated, and 136 quiescence-associated transcripts (gene lists shown in Supplementary Data 2). Only transcripts that showed an RPKM above 2 in our RNA-Seq data were further considered (117 for activation and 124 for quiescence). We filtered this list for genes showing a p(adj) value ≤ 0.05 in the DeSeq2 analysis. Of 117 activation-associated transcripts, 11 were up- and 47 were downregulated in Nf1^Myf5 MPs, including Myog (Supplementary data 2). Of 124 quiescence-associated transcripts, 6 were down- and 78 were upregulated in Nf1^Myf5 MPs (Supplementary data 2). This included Cdkn1b and Cdkn1c encoding cell cycle inhibitors p27 and p57 consistent with cell cycle exit, or Collagen type 5 subunit genes known to be involved in MuSC quiescence[11]. A selection of differentially expressed genes is shown in (Fig. 3e, f). Furthermore, the so-called imprinted gene network, known to be highly expressed in quiescent stem cells[44,45], was upregulated in Nf1^Myf5 MPs (Fig. 3g). We finally compared our RNA-Seq data to the dataset from Ryall et al[39]., comprising 2-month-old MuSCs freshly isolated comparable to our protocol, and MuSCs that were kept for 2 days in culture to reflect activated cells. Comparison of normalized read counts confirmed a shift of Nf1^Myf5 p7 MPs transcriptome toward the signature of quiescent MuSCs (Supplementary Fig. 3f).

RT-qPCR confirmed upregulation of quiescence-related genes Pax7 and Spry1, and downregulation of activation–related genes Myod, Myog, and Myh3 and ATP2a1 (Fig. 3h). Neither Pax7, Calcr, Myog or Myh3 were deregulated in p14 MPs of Nf1-haploinsufficient Myf5^Cre;Nf1^flox/+ mutants (Supplementary Fig. 3g). Freshly isolated Nf1^Myf5 MPs were smaller than control cells (Fig. 3i), a known feature of quiescent MuSCs[46]. In addition, phosphorylation of p70S6 kinase and of S6 ribosomal protein serine-235/236 as mammalian target of

rapamycin complex 1 (mTORC1) signaling readouts, which is known to be induced upon MuSC activation[46], was reduced in Nf1^Myf5 p7 MPs (Fig. 3j, k; full blots for Fig. 3j shown in the source data file).

Results indicated that Nf1^Myf5 p7 MPs show a phenotypic shift toward MuSC quiescence, including a transcriptomic signature, cell cycle exit, and mTORC1 activity.

## Quiescence shift of Nf1^Myf5 MPs reflects an altered epigenetic landscape

We next assessed three major chromatin marks: histone 3 lysine 4 trimethylation (H3K4me3), generally associated with active promoters; H3K27me3, associated with transcription inhibition;[47] and DNA methylation (5'cytosine), globally associated with transcription inhibition[48]. Chromatin immunoprecipitation sequencing (ChIP-Seq) analysis of freshly isolated p7 MPs suggested slightly decreased H3K4me3 levels around the transcriptional start site (TSS) averaged across all genes between controls and Nf1^Myf5 p7 MPs (Fig. 4a), and a global reduction of H3K27me3 levels in Nf1^Myf5 p7 MPs (Fig. 4b). We, however, note that our analysis did not employ IgG or input analysis, or e.g. chromatin spike-in, thus quantitative assumptions at individual loci should be taken with caution. We confirmed globally reduced H3K27me3 levels by immunolabeling (Fig. 4c). GO analysis (biological process) of all 1157 genes with significantly decreased H3K27me3 levels in Nf1^Myf5 p7 MPs identified by DiffBind (Gene list in Supplementary data 3) showed enrichment of terms associated to transcriptional regulation, cellular differentiation, but also "Notch signaling" (Fig. 4d). Intersecting genes with reduced H3K27me3 levels with genes upregulated in Nf1^Myf5 p7 MPs showed only a low overlap of 248 genes (Fig. 4e; gene list in Supplementary data 4), suggesting that reduced H3K27me3 alone cannot explain gene deregulation in Nf1^Myf5 MPs. GO analysis of the 248 genes mostly yielded general terms as "multicellular organism development" or "cell differentiation", but also "transmembrane receptor protein tyrosine kinase signaling pathway" and "positive regulation of kinase activity" in line with increased RAS/MAPK signaling (Supplementary Fig. 4a). However, among these genes were quiescence-associated transcripts Pax7 and Pax3, and Notch pathway components Jag1 and Dll4 (Fig. 4e). ChIP-Seq tracks show decreased apparent H3K27me3 decoration at the Pax7 locus (Fig. 4f).

RNA-Seq analysis showed upregulation of DNA-demethylases Tet1-3 and all three relevant DNA methyltransferases, Dnmt1, Dnmt3a, and Dnmt3b (Fig. 4g). Dnmt1 and Dnmt3a were highly expressed in p7 MPs, but Dnmt3b had low expression levels based on RPKM values (Supplementary Data 1). RT-qPCR confirmed Dnmt1 and Dnmt3a upregulation in Nf1^Myf5 MPs (Fig. 4h). Methylated DNA immunoprecipitation sequencing (MeDIP-Seq) analysis of freshly isolated p7 MPs showed differential methylation between controls and Nf1^Myf5 predominantly at CpG islands (Fig. 4i). Enrichment analysis of differentially methylated regions (DMRs) showed that predominantly CpG islands gained methylation in Nf1^Myf5 MPs (Fig. 4i).

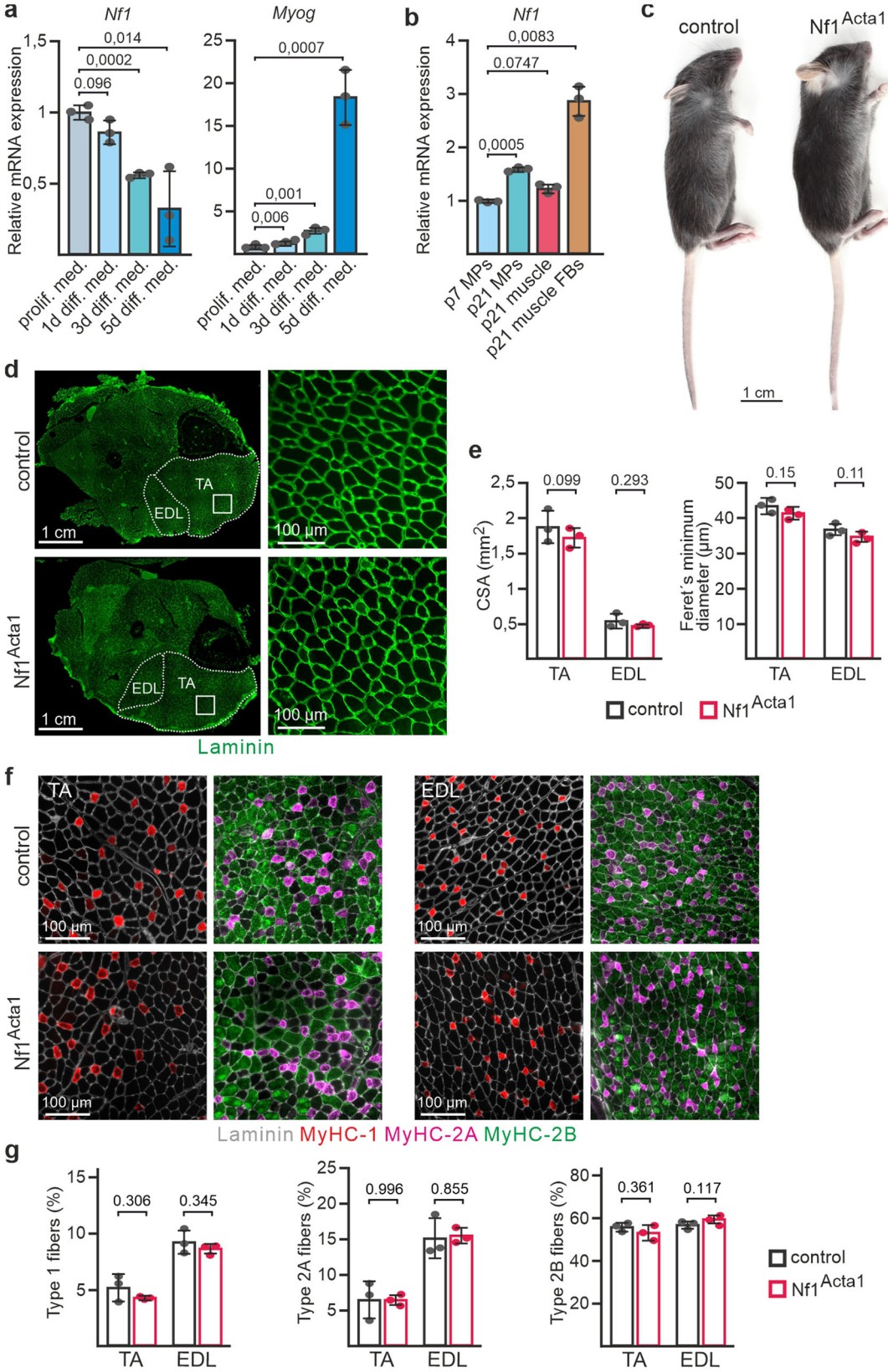

Gene Ontology (GO) overrepresentation analysis of genes in proximity to CpG islands with increased methylation in Nf1$^{Myf5}$ MPs showed terms associated with RNA synthesis and transcription, and cellular metabolism (Fig. 4j). Mining the proximity of differentially methylated regions for myogenesis-related genes showed a CpG island with gain of methylation in Nf1$^{Myf5}$ MPs 3.5 kilobases (kb) upstream of *Myl1* (Fig. 4k), which is part of the activation signature and is downregulated in Nf1$^{Myf5}$ MPs in transcriptome data (Fig. 4l). RT-qPCR confirmed *Myl1* downregulation (Fig. 4m). Analysis of metabolism-related genes showed gain of methylation of a CpG island 1.5 kb upstream of *Pfkfb1* in Nf1$^{Myf5}$ MPs, encoding phosphofructo-kinase-fructose-bisphosphatase 1 (Fig. 4n). *Pfkfb1* was the highest expressed isoform

**Fig. 2 | Nf1 is dispensable in myofibers. a** Quantitative real-time PCR for *Nf1* (left) and *Myog* (right) on primary mouse myoblasts cultured in proliferation medium or upon myogenic induction for indicated time (*n* = 3 animals per genotype; each dot represents the mean of three technical replicates from one biological replicate; *p*-values shown). **b** Quantitative real-time PCR for *Nf1* in p7 or p21 MPs, p21 whole muscle tissue or p21 muscle-derived fast-adhering fibroblastic cells (FBs) (*n* = 3 animals per genotype; each dot represents the mean of three technical replicates from one biological replicate; *p*-values shown). **c** Whole-body appearance of control and Nf1[Acta] mice at p21. **d** Cross sections of lower hind limbs of control and Nf1[Acta] mice at p21 immunolabeled for Laminin (green), TA Tibialis anterior, EDL

Extensor digitorum longus. Magnifications of indicated areas in TA muscles shown right. **e** Quantification of cross-sectional area (CSA; left) and myofiber Feret's minimum diameter (right) of control and Nf1[Acta] TA and EDL muscles (*n* = 3 animals per genotype; *p*-values shown). **f** Immunolabeling for Laminin (gray), MyHC-1 (red), MyHC-2A (purple), and MyHC-2B (green) on cross sections of TA (left) and EDL (right) muscles of control and Nf1[Acta] mice at p21. **g** Quantification of fiber types in p21 control and Nf1[Acta] TA and EDL muscles (*n* = 3 animals per genotype; *p*-values shown). Data are mean ± SEM; *P*-value calculated by two-sided unpaired *t*-test. Source data are provided as a Source Data file.

---

of all *Pfkfb*s in juvenile MPs (Supplementary Data 1), and *Pfkfb1* mRNA expression was strongly downregulated in Nf1[Myf5] MPs in transcriptome data (Fig. 4o) and RT-qPCR (Fig. 4p). In addition, gain of methylation in Nf1[Myf5] MPs overlapped the promoter of *Ndufb11*, encoding a subunit of mitochondrial complex I, and *Idh3g*, encoding a subunit of mitochondrial isocitrate dehydrogenase, which catalyzes the rate-limiting step of the tricarboxic acid cycle (TCA) (Supplementary Fig. 4b).

Therefore, epigenetic alterations in Nf1[Myf5] MPs, at the genes we subjected to validation, are consistent with a shift toward quiescence. These alterations could contribute to impaired myogenic differentiation, and gain of methylation and transcriptional downregulation of metabolic genes indicate changes in cellular energy metabolism in Nf1[Myf5] MPs.

## Metabolic reprogramming of Nf1[Myf5] juvenile MPs that is conserved in myofibers

Consistent with possible metabolic alterations, overrepresentation analysis with KEGG database pathways of the p7 MP transcriptome showed enrichment of "metabolic pathways," "carbon metabolism," "biosynthesis of amino acids," and "glycolysis/gluconeogenesis" in genes downregulated in Nf1[Myf5] MPs, and "protein digestion and absorption" in upregulated genes (Supplementary Fig. 3e). GSEA showed enrichment of "glycolysis/gluconeogenesis" and "oxidative phosphorylation" in controls (Fig. 5a). We identified global downregulation of genes of the glycolytic pathway and the pyruvate dehydrogenase complex, the citrate cycle, and the mitochondrial electron transport chain (ETC) in Nf1[Myf5] p7 MPs (Fig. 5b–d). Expression of glycolytic genes *Hk2* and *Pfkfb1* was unaltered in p14 MPs of *Nf1*-haploinsufficient Myf5[Cre];Nf1[flox/+] mutants (Supplementary Fig. 5a).

Analyzing transcriptomic data by kinetic metabolic modeling[49,50] confirmed a widespread metabolic shutdown in Nf1[Myf5] MPs decreasing uptake and utilization of glucose, fatty acids, and branched-chain amino acids accompanied by decreased capacity for ATP production and oxygen consumption (Supplementary Fig. 5b).

Seahorse real-time metabolic flux analysis with freshly isolated p7 MPs to assess the metabolic consequence of this deregulation showed a strong reduction of extracellular acidification rate (ECAR) in Nf1[Myf5] MPs, indicating severe glycolytic flux inhibition (Fig. 5e). In contrast, Nf1[Myf5] p7 MPs showed only a moderate decrease in the basal oxygen consumption rate (OCR) below statistical significance in (Fig. 5f), indicating that oxidative phosphorylation capacity is still sufficient to maintain resting energy demand in MPs. Therefore, Nf1[Myf5] MPs mainly use low-level oxidative metabolism as an energy source, consistent with a quiescent phenotype[51].

Differentiated muscle in Nf1[Myf5] animals showed fast fiber atrophy, shift of glycolytic to oxidative fiber types, and increased oxygen consumption[31]. This was confirmed by kinetic metabolic modeling analysis of p21 Nf1[Myf5] muscle tissue transcriptome data[31] suggesting impaired capacity for glucose utilization, but increased capacity for fatty acid utilization and unchanged capacity for utilization of branched-chain amino acids, concomitant with increased capacity for ATP production and increased oxygen consumption (Supplementary Fig. 5c). Direct comparison of Nf1[Myf5] p7 MP transcriptome data to Nf1[Myf5] p21 muscle data showed common downregulation of only 130

genes (Fig. 5j). GO overrepresentation analysis of this gene set showed enrichment of terms related to glucose/carbon metabolism and amino acid synthesis (Fig. 5g). GO analysis of 799 genes downregulated in p21 Nf1[Myf5] muscle, but not in p7 MPs, did not yield any metabolism-related terms (Supplementary Fig. 5d), but rather terms as "Z-disc" possibly reflecting fiber atrophy[31]. Thus, transcriptome analysis and metabolic flux modeling of Nf1-deficient progenitors and differentiated muscle indicate a continuous deregulation of specifically carbohydrate metabolism. As Nf1[Acta] mice showed no significant defect in muscle size, fiber types, and expression of a panel of metabolic genes, this suggests that perturbed muscle fiber metabolism in Nf1[Myf5] animals[31] can be traced back to defects in juvenile MPs, indicating that that metabolic reprogramming in Nf1[Myf5] juvenile MPs is transmitted to myofibers.

A metabolic switch from slow oxidative to forced glycolytic metabolism occurs during adult MuSC exit from quiescence to activation[39]. Concomitant NAD[+] depletion inhibits Sirt1 function, which acts as a histone deacetylase mainly targeting H4K16. Increased H4K16 acetylation upon muscle-specific Sirt1 deletion induces expression of MuSC activation and myogenic differentiation-related genes[39]. H4K16ac ChIP-Seq showed a decrease in global levels of H4K16ac in p7 Nf1[Myf5] MPs (Fig. 5h). Analysis of global H4K16ac levels in freshly isolated p7 MPs (Fig. 5i) and in p7 and 12-week-old muscle sections by immunofluorescence (Supplementary Fig. 6a, b) confirmed long-term decreased H4K16ac levels in Nf1[Myf5] Pax7[+] cells compared to controls.

*Myh3*, *Bgn*, *Fst* and *Mylk2*, which are upregulated in MuSC-specific Sirt1 conditional mice[39], were downregulated in Nf1[Myf5] MPs (Supplementary Fig. 6c) and showed reduced H4K16ac decoration at their gene bodies (Fig. 5j and Supplementary Fig. 6d). Thus, Nf1[Myf5] MPs are driven toward quiescence and show metabolic reprogramming with severely inhibited glycolytic metabolism, and decreased H4K14ac and expression of myogenic differentiation-related genes.

## Increased Notch signaling induced by a Mek/Erk/NOS cascade drives Nf1[Myf5] juvenile MP quiescence shift

To analyze the mechanism of Nf1[Myf5] MPs quiescence shift and reprogramming, we performed in vitro culture of juvenile primary myoblasts. RT-qPCR confirmed effective *Nf1* knockdown (Supplementary Fig. 7a). Surprisingly, in vitro, Nf1[Myf5] myoblasts did not reproduce the in vivo proliferative behavior, but showed enhanced proliferation (Supplementary Fig. 7b). Switching cells to a differentiation medium after 2 d of culture showed effective block of myogenic differentiation in Nf1[Myf5] primary MPs (Supplementary Fig. 7b), as observed before for FACS-isolated MPs. Both increased proliferation and blocked differentiation fully depended on Mek/Erk signaling, as shown by inhibition with UO126 (Supplementary Fig. 7b). The discrepancy between in vivo and in vitro proliferation behavior of Nf1[Myf5] MPs indicated that in vivo non-cell-autonomous microenvironmental factors override or divert Mek/Erk signaling in juvenile MPs, inhibiting cell proliferation.

Transcriptome analysis indicated upregulation of Delta/Notch signaling pathway components *Dll1* and *Notch1/3*, and upregulation of Notch pathway targets *Hes1*, *Hey1*, *Heyl*, *Calcr*, *Col5a1* and *Col5a3* (Fig. 3f). GSEA showed enrichment of "Notch targets" in Nf1[Myf5] MPs (Supplementary Fig. 7c). Upregulation of *Notch1*, *Notch3*, *Hes1*, and

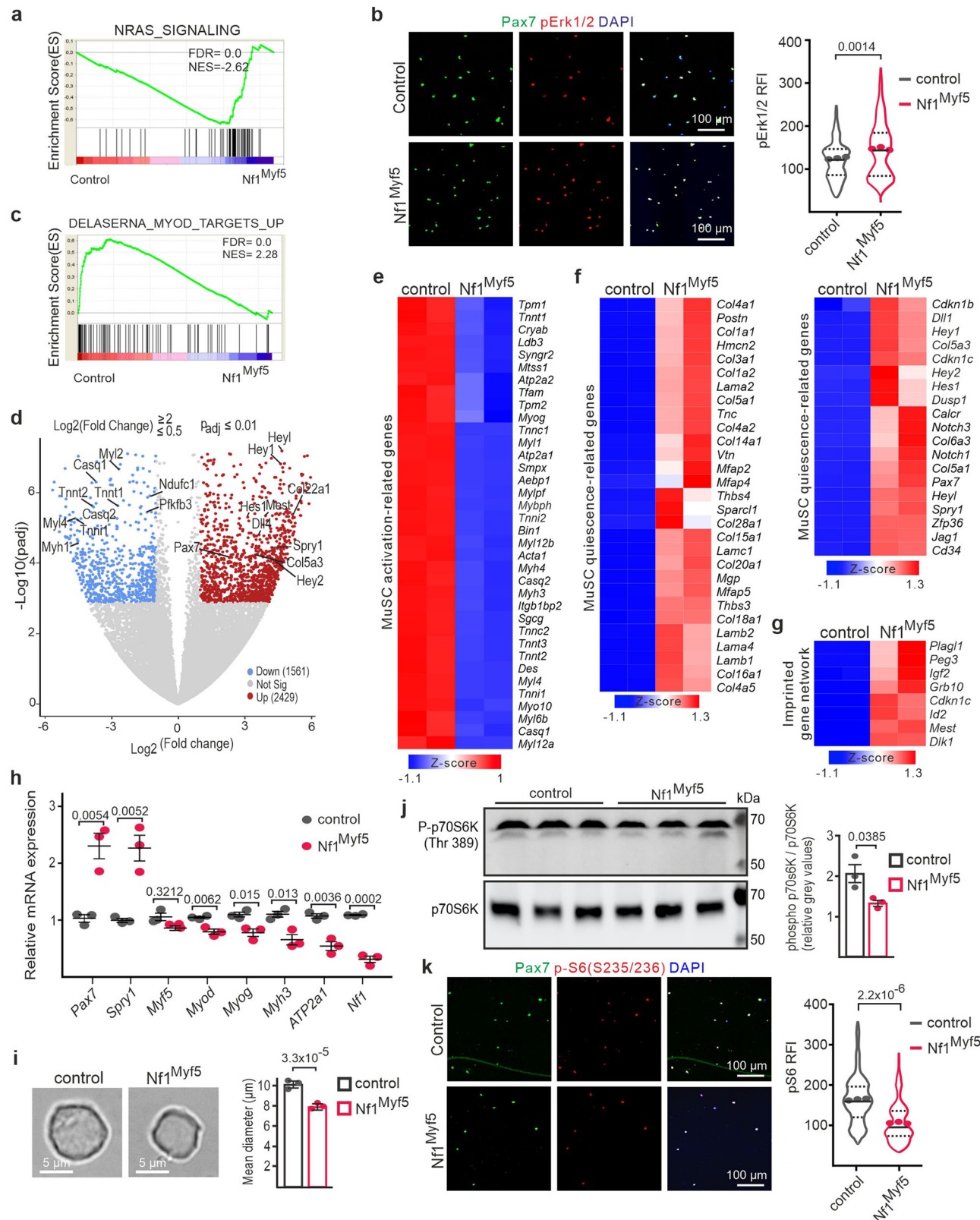

*Hey1* in p7 Nf1^Myf5 MPs was confirmed by RT-qPCR (Fig. 6a). *Hes1*, *Hey1* and *Notch1* expression was unaltered in p14 MPs of *Nf1*-haploinsufficient Myf5^Cre;Nf1^flox/+ mutants (Supplementary Fig. 7d).

We therefore tested whether Notch signaling represents the in vivo niche factor lacking in vitro by culturing MPs on cell culture plates coated with recombinant Jagged-1, which activates Notch signaling in myogenic cells[52–54]. To calibrate the system, we first cultured wild type primary myoblasts for 2 d in proliferation medium on uncoated control dishes, or dishes coated with different concentrations of Jagged-1 ligand. This showed induction of *Hes1* and *Hey1* expression already at 2.5 ng/µl, with maximal induction reached at 5 ng/µl (Supplementary Fig. 7e). Compared to control cells, Nf1^Myf5 primary myoblasts showed increased induction of Notch target gene expression on recombinant Jagged-1–coated (5 ng/µl) dishes (Fig. 6b), indicating that Nf1^Myf5 MPs

**Fig. 3 | Premature shift of Nf1^Myf5 MPs to quiescence. a** GSEA of control and Nf1^Myf5 p7 MP RNA-Seq data for "NRAS Signaling". **b** Labeling of cytospun control and Nf1^Myf5 p7 MPs for Pax7 (green), phosphor-ERK1/2 (pErk1/2, red) and DAPI (nuclei, blue). Quantification of relative fluorescence intensity (RFI) for anti-pErk1/2 shown right. Data range is shown as violin plot with median and interquartile range, means of biological replicates are shown as dots (*n* = 3 animals per genotype; *p*-value shown). **c** GSEA of RNA-Seq data from control or Nf1^Myf5 p7 MPs shows "MyoD targets" enriched in controls. **d** Volcano plot of transcriptome data from freshly FACS-isolated control or Nf1^Myf5 p7 MPs. Individual transcripts deregulated in Nf1^Myf5 MPs are indicated (blue: down; red: up). DE genes were identified by a log2 fold change over 2 or below 0.5 and a Benjamini-Hochberg adjusted *p*-value (padj) <0.01. Only genes with RPKM above 2 were considered. **e** Heatmap shows reduced MuSC activation–related gene expression in p7 Nf1^Myf5 MPs. **f** Heatmap shows increased MuSC quiescence-related gene expression in p7 Nf1^Myf5 MPs. **g** Heatmap shows increased expression of imprinted gene network genes in p7 Nf1^Myf5 MPs. **h** RT-qPCR confirmation of differential expression of indicated genes in Nf1^Myf5 p7 MPs (*n* = 3 animals per genotype; each dot represents the mean of three technical replicates from one biological replicate; *p*-values shown). **i** Reduced cell diameter in p7 Nf1^Myf5 freshly sorted MPs. Representative images (left); quantification (right) (*n* = 3 animals per genotype; *p*-value shown). **j** Western blot shows reduced p70s6 kinase phosphorylation at Thr-389 in p7 Nf1^Myf5 MPs (*n* = 3 animals per genotype; *p*-value shown). **k** Labeling of cytospun control and Nf1^Myf5 p7 MPs for Pax7 (green), phosphor-Serine-235/236 S6 ribosomal protein (p-S6, red) and DAPI (nuclei, blue). Quantification of relative fluorescence intensity (RFI) for anti-p-S6 shown right. Data range is shown as violin plot with median and interquartile range, means of biological replicates are shown as dots (*n* = 3 animals per genotype; *p*-values shown). Data are mean ± SEM; *P*-value calculated by two-sided unpaired *t*-test. Source data are provided as a Source Data file.

are hypersensitive to Notch pathway activation upon external ligand stimulation.

Placement of Nf1^Myf5 p14 FACS-isolated MPs cultured for 2 d in proliferation medium on uncoated dishes showed increased proliferation (Fig. 6c, d), as seen for primary myoblasts before (Supplementary Fig. 7b). We thus assessed, whether Jagged-1 treatment could reduce MP proliferation. Cultivation on Jagged-1-coated dishes (5 ng/μl) led to a reduced proliferation rate already in control p14 MPs, which was exacerbated in Nf1^Myf5 MPs (Fig. 6c, d). This was confirmed by a dose-response titration using primary myoblasts showing that 2.5 ng/μl Jagged-1 reduced Nf1^Myf5 myoblast proliferation rates to control levels, while higher concentrations reduced Nf1^Myf5 myoblast proliferation rates below control levels (Supplementary Fig. 7f).

Both control and Nf1^Myf5 MPs had low Pax7 expression after 2 d cultivation on control dishes (Fig. 6c, e) consistent with previous observations[55]. Jagged-1 maintained Pax7 expression in control and Nf1^Myf5 MPs (Fig. 6c, e), with a relative increase in Pax7 abundance in Nf1^Myf5 MPs (Fig. 6e). Conversely, Jagged-1 reduced the relative numbers of MyoD^+ cells (Fig. 6f, g) and MyoD abundance (Fig. 6f, h) in control MPs, which was both exacerbated in Nf1^Myf5 MPs (Fig. 6f–h), indicating that Jagged-1 induces a shift toward quiescence in juvenile control MPs, which is intensified in Nf1^Myf5 MPs.

In Nf1-deficient oligodendrocytes, a Mek/Erk/nitric oxide synthase (NOS)/cyclic guanosine monophosphate (cGMP)/protein kinase G (PKG) pathway drives Notch pathway activation[56]. GSEA showed NO-cGMP signaling enriched in Nf1^Myf5 MPs (Fig. 6i). Consistent with this, Nf1^Myf5 MPs placed on Jagged-1-coated dishes and treated with Mek inhibitor UO126 or pan-NOS inhibitor L-NAME canceled the hyperresponsiveness to Jagged-1 (Fig. 6j), although we cannot formally exclude an effect of the inhibitors independent of Jagged-1 treatment. We conclude that in juvenile MPs, a Ras/Mek/Erk/NOS pathway funnels into activation of the Notch pathway, inducing quiescence, which is exacerbated by lack on Nf1.

### Inhibition of Notch signaling prevents quiescence shift of Nf1^Myf5 juvenile MPs and ameliorates the whole-body phenotype of Nf1^Myf5 mice

Notch signaling regulates cell metabolism in several systems[57–59]. We thus first analyzed whether Notch signaling is upstream of metabolic gene expression in juvenile MPs. RT-qPCR of selected glycolysis and mitochondrial gene expression levels in WT p14 MPs cultured on control dishes or in the presence of Jagged-1 indicated that the Notch pathway can inhibit energy metabolism-related gene expression in juvenile MPs (Fig. 7a). Jagged-1 stimulation especially affected glycolytic genes as *Pfkfb1*, *Pfkfb3*, *Pfkm*, *Eno3*, *Ldha* and *Hk2*, and it mildly affected *mtCO1* and *Ndufv1* as representatives of the TCA cycle and ETC (Fig. 7a), overlapping transcriptome data of Nf1^Myf5 p7 MPs. This suggests that in juvenile MPs, activation of the Notch pathway contributes to metabolic reprogramming by inhibiting glycolytic gene expression.

To test whether Notch signaling is needed for premature quiescence induction and long-term metabolic reprogramming in vivo, we treated Nf1^Myf5 pups with 5 doses of 30 mg/kg of the Notch pathway inhibitor DAPT (or placebo control) from p6 to p18 (Fig. 7b). DAPT is an inhibitor of γ-Secretase, preventing Notch cleavage and thus signal transduction to the nucleus. To monitor Notch pathway inhibition in vivo, we measured the expression of two Notch pathway components and four Notch pathway targets using RT-qPCR and found reduced expression of all genes analyzed (Supplementary Fig. 8). In vivo DAPT treatment increased Pax7^+ cell numbers (Fig. 7c, d) and Pax7^+ cell proliferation (Fig. 7c, e) compared to placebo-treated Nf1^Myf5 mice. Thus, premature quiescence induction was prevented by in vivo DAPT treatment. In vivo DAPT treatment increased expression of glycolytic genes in p21 Nf1^Myf5 muscle, while *Ndufv1* and *mtCO1* expression stayed the same (Fig. 7f).

Nf1^Myf5 mice show muscle atrophy and a whole-body catabolic phenotype with attrition of white adipose tissue because of increased muscular consumption of fatty acids[31]. In vivo DAPT treatment increased the apparent myofiber size (Fig. 7g) and the body weight (Fig. 7h) of Nf1^Myf5 mice, indicating partial rescue of the muscle growth phenotype. In addition, in vivo DAPT treatment of Nf1^Myf5 mice increased the white adipose tissue depot weight (Fig. 7i), indicating rescue of aberrant myofiber lipid metabolism.

We conclude that the Notch pathway is needed in vivo to induce premature quiescence in Nf1^Myf5 MPs, and inhibition of the Notch pathway ameliorates metabolic reprogramming and improves the whole-body catabolic phenotype of Nf1^Myf5 mice.

## Discussion
Early postnatal myofiber growth depends on a combination of cell accrual and metabolic growth. In addition to reduced metabolic growth[31], Nf1^Myf5 muscle shows highly reduced myonuclear accretion during early postnatal growth.

### Nf1 counteracts Notch-dependent postnatal quiescence induction
To enable postnatal muscle growth by myonuclear accretion, a Pax7^+ MP pool is maintained in proliferation, from which cells continuously differentiate and fuse to preexisting fibers. This, however, needs to be coordinated with quiescence induction in a subset of cells, which is required to establish a permanent MuSC pool. In Nf1^Myf5 mice, postnatal MPs show premature quiescence induction evidenced by precocious and lasting cell cycle exit, reduced differentiation capability, reduced cell size and mTORC1 activity, and a transcriptomic signature showing increased expression of quiescence-related and reduced expression of differentiation-related genes. Therefore, the intricate balance between cell cycle exit/quiescence induction and maintenance of the progenitor pool and concomitant differentiation is distorted, leading to precocious contraction of the Pax7^+ pool and so decreasing the numbers of both, myonuclei and MuSCs.

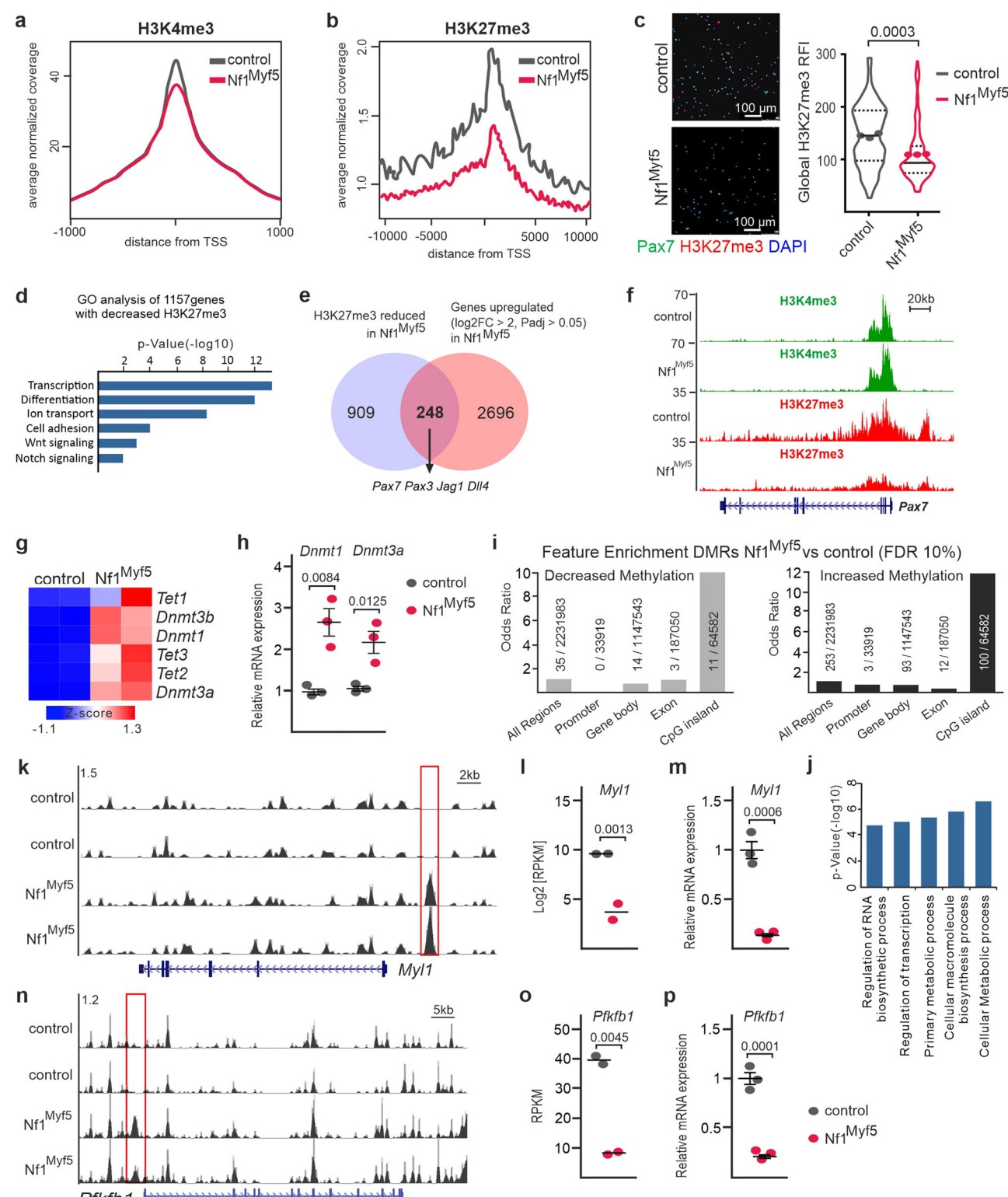

Quiescence induction is concomitant with changes in the epigenetic landscape. Reduced global H3K27me3 levels in Nf1^Myf5 MPs overlap reduced H3K27me3 levels in quiescent adult MuSCs[43,55]. Reduced H3K27me3 levels at the Pax7 locus and consistently increased mRNA and protein expression might support a shift to quiescence[43,60]. Global H3K4me3 levels were not changed in Nf1^Myf5 MPs, also resembling adult MuSCs[55]. However, a slight reduction of H3K4me3 at TSS across all genes was seen; early activation of MuSCs causes an increase in H3K4me3 levels at myogenic loci, including *Mylk2*[43], while in Nf1^Myf5

MPs, this locus showed reduced H3K4me3 decoration (Supplementary Fig. 6d).

Nf1^Myf5 juvenile MPs show an increase in methylated DNA regions, including myogenic genes such as *Myl1*. The gain of methylation is consistent with previous studies showing increased DNA methylation in quiescent MuSCs[61,62] and increased heterochromatin content of quiescent MuSCs[63,64]. This suggests that postnatal quiescence induction follows a change in the epigenetic landscape that contrasts adult MuSC quiescence exit.

**Fig. 4 | Epigenetic changes associated with quiescence shift of Nf1$^{Myf5}$ MPs.**
**a, b** Averaged normalized coverage in given region surrounding the transcriptional start site (TSS) across all genes for H3K4me3 and H3K27me3, derived from ChIP-Seq performed on control and Nf1$^{Myf5}$ FACS-isolated p7 MPs. **c** Immunolabeling for Pax7 (green) and H3K27me3 (red) on cytospun control and Nf1$^{Myf5}$ p7 MPs. Quantification of anti-H3K27me3 relative fluorescence intensity (RFI) is shown right. Data range is shown as violin plot with median and interquartile range, means of biological replicates are shown as dots (*n* = 3 animals per genotype; *p*-values shown). **d** GO analysis of all genes with significantly reduced H3M27me3. **e** Intersection of genes with reduced H3K27me3 and genes upregulated in p7 Nf1$^{Myf5}$ MPs. **f** ChIP-Seq tracks for H3K4me3 and H3K27me3 at the *Pax7* locus in control and Nf1$^{Myf5}$ p7 MPs. **g** Heatmap depiction of DNA methylation–related gene expression in control and Nf1$^{Myf5}$ p7 MPs. **h** RT-qPCR of *Dnmt1* and *Dnmt3a* in control and Nf1$^{Myf5}$ p7 MPs (*n* = 3 animals per genotype; *p*-values shown). **i** Enrichment analysis of Regions with increased or decreased methylation levels in Nf1$^{Myf5}$ vs. control p7 MPs for different regions of interest (ROIs; promoter defined TSS as ±500 bases). Bar height corresponds to the odds ratio of DMIs in specific ROIs (ratios of numbers of DMRs in specific ROIs relative to the ratios of numbers of DMRs found in all regions analyzed). **j** GO analysis of regions with increased or decreased methylation levels in Nf1$^{Myf5}$ vs. control p7 MPs. **k** MeDIP-Seq tracks from control and Nf1$^{Myf5}$ p7 MPs at the *Myl1* locus. **l** Log2(RPKM) values for *Myl1* in p7 MPs transcriptome data (*n* = 2 animals per genotype; mean values and Padj.-value shown). **m** RT-qPCR of *Myl1* expression in control and Nf1$^{Myf5}$ p7 MPs (*n* = 3 animals per genotype; *p*-value shown). **n** MeDIP-Seq tracks from control and Nf1$^{Myf5}$ p7 MPs at the *Pfkfb1* locus. **o** RPKM values for *Pfkfb1* in p7 MP transcriptome data (*n* = 2 animals per genotype; mean values and Padj.-value shown). **p** RT-qPCR of *Pfkfb1* expression in control and Nf1$^{Myf5}$ p7 MPs (*n* = 3 animals per genotype; *p*-value shown). Data are mean ± SEM; *P*-value calculated by two-sided unpaired *t*-test. Source data are provided as a Source Data file.

Deficiency of tumor suppressors such as *Nf1* is usually associated with increased cell proliferation[23]. Opposite to their in vivo behavior, Nf1$^{Myf5}$ MPs show enforced proliferation in vitro, which fully depends on Erk signaling. In accordance, Mek/Erk signaling is needed for in vitro myoblast proliferation[65], and Erk promotes MuSC proliferation in isolated myofibers[66]. In contrast, Erk1/2 signaling has also been involved in quiescence induction in myoblasts in vitro;[67] Mek/Erk signaling downstream of Ang-1/Tie2 induces quiescence-related gene expression in MPs in vitro, and in vivo Ang-1 overexpression increases and Tie2 blockade reduces the number of quiescent MuSCs[15]. This altogether indicates that Mitogen-activated protein kinase (Mapk/Erk) signaling in MPs/MuSCs is time- and context-dependent and might strongly depend on the local microenvironment.

Notch pathway activation via recombinant Jagged-1 completely reverses in vitro hyperproliferation of Nf1$^{Myf5}$ juvenile MPs, and Jagged-1 stimulation of both Nf1$^{Myf5}$ and WT MPs increases Pax7 expression, reduces proliferation, and reduces MyoD expression, thus recapitulating the in vivo behavior. Inhibition of Notch signaling in vivo prevents precocious cell cycle exit and rescues muscle growth and MuSC numbers in Nf1$^{Myf5}$ mice. Although in this scenario we cannot exclude that Notch inhibition affects other cells apart from MPs, the outcome is in line with the cell-autonomous effects observed in vitro. Nf1$^{Myf5}$ MPs are hypersensitive to Notch ligand stimulation, indicating a synergistic action of Ras/Mek/Erk and Notch signaling in early postnatal MPs, which is also observed in other contexts[68,69]. As in oligodendrocytes[56], Nf1/Ras/Mek/Erk and Notch pathways are interconnected in juvenile MPs by a conserved NOS-dependent pathway. Notch signaling is essential for the maintenance of adult satellite cell quiescence[12,14,18,21], and is involved in postnatal MPs quiescence induction[20,21,70] as well as return of injury-activated MuSCs to quiescence, thereby promoting cell cycle exit and Pax7 expression[17], which is associated with deeper quiescence[71]. Our data further support the role of Notch signaling in driving postnatal MP quiescence at the expense of transient amplifying pool amplification and differentiation. These findings indicate an intrinsic function for Nf1 controlling Ras/Mek/Erk-based Notch pathway intensification in juvenile MPs, preventing quiescence induction, allowing pool expansion, and safeguarding postnatal muscle growth.

## Metabolic reprogramming in juvenile MPs quiescence induction
KEGG analysis of Nf1$^{Myf5}$ MP transcriptome data indicates a global downregulation of anabolic processes, and GO analysis of regions with gain of methylation in Nf1$^{Myf5}$ MPs suggests that biosynthetic processes, RNA synthesis, and transcription are affected. This result is consistent with a global decrease in biosynthesis and transcriptional activity in quiescent stem cells[72] and suggests the induction of a transcriptionally controlled state of metabolic quiescence in Nf1$^{Myf5}$ MPs. Adult MuSC activation is associated with an increase in glycolytic metabolism[39,43,51,73], which is also seen in several other stem cell types, as glycolysis satisfies the proliferating cell's demand for quick energy production and metabolite supply[74,75]. Most glycolytic genes are downregulated in Nf1$^{Myf5}$ juvenile MPs, consistent with a highly reduced ECAR. Of these, *Pfkfb1* encodes one of four 6-phosphofructo-2-kinase/fructose-2,6-biphosphatase enzymes and is a key positive regulator of glycolytic flux[76]. Together with the downregulation of genes encoding other key glycolysis pacemakers, such as *Hk2*, *Pfkm*, *Pkm*, and genes of the pyruvate dehydrogenase complex, this may explain the stalled glycolysis in Nf1$^{Myf5}$ juvenile MPs. Notably, also genes of other carbohydrate metabolism pathways, such as the serine-one carbon and pentose phosphate pathways (*Phgdh, Psat1, Psph, G6pdx*; Supplementary Data 1) are downregulated, suggesting reduced folate metabolism and pentose production, both needed for nucleotide biosynthesis and cell division. Oxidative metabolism is, however, only mildly affected in p7 Nf1$^{Myf5}$ MPs. Genes encoding ETC subunits or citrate cycle enzymes are mildly downregulated, and genes for fatty acid metabolism are mostly unaffected. This is consistent with the idea that quiescent MuSCs mainly depend on low-level fatty acid–driven oxidative phosphorylation (OXPHOS)[73].

Notch signaling is highly context-dependent and promotes glycolysis, for example, in hematopoietic stem cells or breast cancer cells[77,78] or represses glucose metabolism, for example, in the developing nervous system[79] or mesenchymal stem cells[80]. Treatment of WT juvenile MPs with a Notch ligand strongly inhibits glycolytic gene expression and to a milder extent OXPHOS gene expression, indicating a direct role of Notch signaling in metabolic quiescence induction in MPs.

Reduced glycolytic flux increases cellular NAD$^+$ levels, which activate Sirt1 in MuSCs, leading to deacetylation and inhibition of myogenic differentiation genes[39]. H4K16ac decoration was reduced in Nf1$^{Myf5}$ juvenile MPs. Myogenic genes affected in Nf1$^{Myf5}$ MPs overlap genes identified as Sirt1 targets in MuSCs[39], thus it is possible that Sirt1-mediated metabolic reprogramming of Nf1$^{Myf5}$ juvenile MPs may contribute to impairing their myogenic differentiation. Therefore, inhibition of glycolytic genes downstream of Notch signaling, which is exacerbated by Ras/Mek/Erk input in Nf1$^{Myf5}$ MPs, might be key for driving postnatal MP quiescence via metabolic reprogramming.

The main features of metabolic reprogramming are conserved in fully differentiated Nf1$^{Myf5}$ muscle on a transcriptome level and functionally, with stalled glycolytic metabolism resulting in an energy deficit and muscle atrophy[31]. Myofiber-specific *Nf1* inactivation has no obvious deleterious consequences, indicating that metabolic perturbation of Nf1$^{Myf5}$ muscle fibers is secondary to Nf1/Notch-mediated reprogramming of juvenile MPs, manifesting in long-term inhibition of glycolytic genes and persistently inhibiting carbohydrate metabolism.

NF1 is an autosomal-dominant disorder featuring systemic Nf1 haploinsufficiency; however certain features such as neurofibromas or pseudarthroses can harbor loss-of-heterozygosity (LOH)[81]. Analysis of NF1 protein in human NF1 muscle samples showed an approximate 50% reduction in line with haploinsufficiency[32], however LOH in only

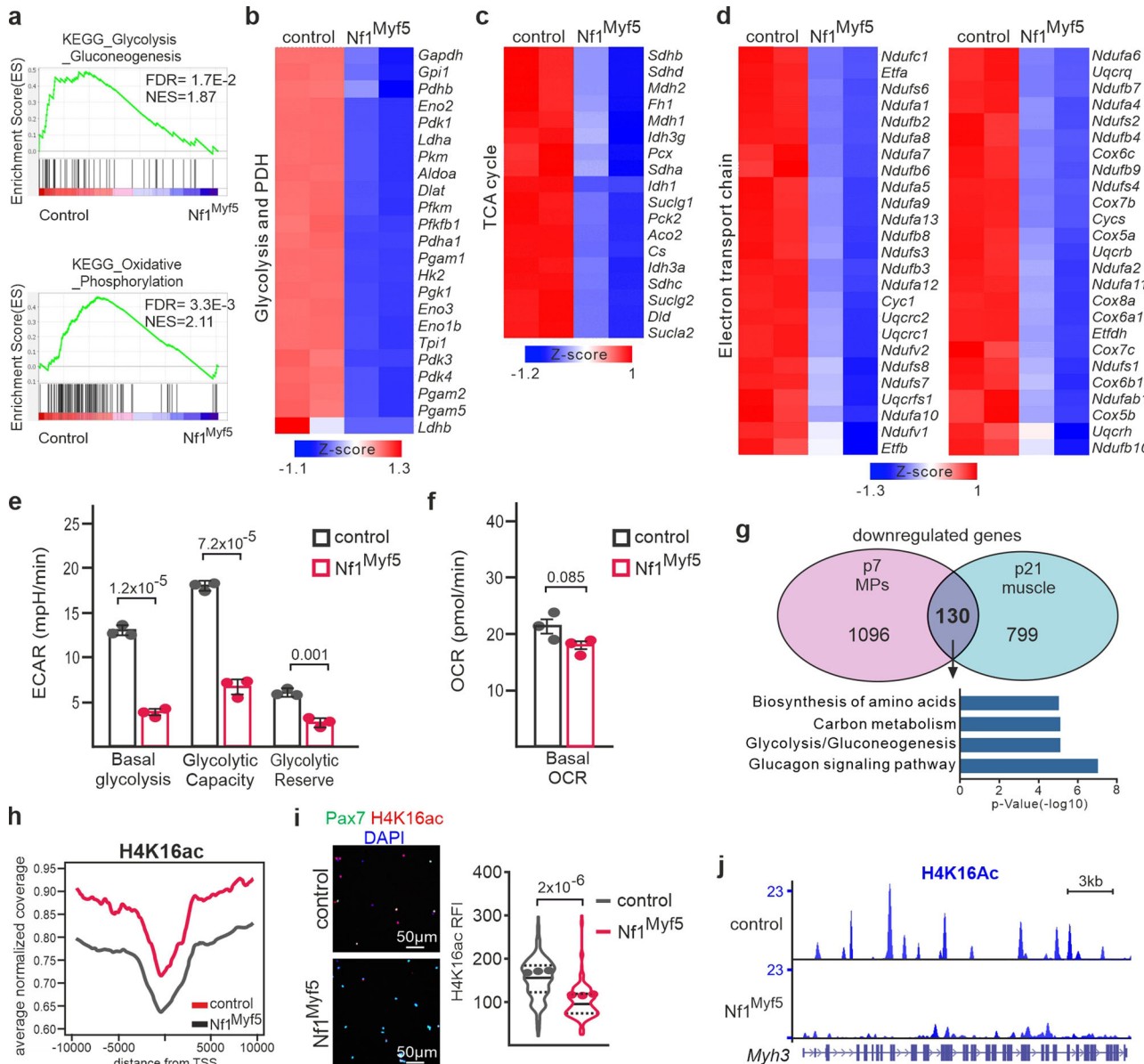

**Fig. 5 | Metabolic reprogramming of Nf1^Myf5 MPs. a** GSEA of control and Nf1^Myf5 p7 MP RNA-Seq data for "glycolysis - gluconeogenesis" and "oxidative phosphorylation." **b–d** Heatmaps show significant DEGs in Nf1^Myf5 versus control MPs related to glycolysis and pyruvate dehydrogenase complex (**b**), TCA cycle components (**c**) and electron transport chain components (**d**). **e** SeahorseXF flux analysis of control and Nf1^Myf5 p7 MPs; quantification of ECAR (n = 3 independent biological replicates from 3 animals per genotype; each dot represents the mean of five technical replicates from one biological replicate; p-values shown). **f** SeahorseXF flux analysis of control and Nf1^Myf5 p7 MPs; quantification of OCR (n = 3 independent biological replicates from 3 animals per genotype; each dot represents the mean of five technical replicates from one biological replicate; p-values shown). **g** Venn diagram showing 130 commonly downregulated genes between Nf1^Myf5 p7 MPs and Nf1^Myf5 p21 muscle. GO analysis of commonly downregulated genes shown below. **h** Averaged normalized coverage for H4K16ac derived from ChIP-Seq on control and Nf1^Myf5 p7 FACS-isolated MPs. TSS, transcription start site. **i** Immunolabeling for Pax7 (green) and H4K16ac (red) on FACS-isolated cytospun MPs from p7 control and Nf1^Myf5 animals. Quantification of anti-H4K16ac relative fluorescence intensity (RFI) is shown right. Data range is shown as violin plot with median and inter-quartile range, means of biological replicates are shown as dots (n = 3 animals per genotype; p-value shown). **j** ChIP-Seq tracks for H4K16ac from control and Nf1^Myf5 p7 MPs at the *Myh3* locus. Data are mean ± SEM; P-value calculated by two-sided unpaired t-test. Source data are provided as a Source Data file.

specific cell type(s) of muscle tissue cannot be excluded. Neither muscle-haploinsufficient Myf5^Cre;Nf1^flox/+ mice (this study) nor Nf1^+/- ± mice[29] show a discernable muscle phenotype. Overall, Nf1-haploinsufficient mice only in part recapitulate the spectrum of human NF1 haploinsufficiency-related features[81], requesting homozygous inactivation of Nf1 to model disease manifestations, even if in humans these are not associated with LOH[29,82]. Indeed, our data suggest a compensatory upregulation of *Nf1* expression from the intact allele in MPs (this study), and muscle tissue[31] in mice. Nevertheless, while a homozygous Nf1 inactivation in mouse model may not recapitulate the

exact genetic background seen in NF1 patients, this model allows to dissect the function of Nf1 in a specific tissue type, in this case muscle and myogenic progenitors.

Thus, in a muscle-specific NF1 mouse model, the muscular metabolic phenotype can be traced back to *Nf1* activity in juvenile MPs. This suggests transmission of differential gene expression signatures across cell divisions and cellular differentiation, which might involve epigenetic memory[83]. Why in our model specifically glycolytic genes are continuously repressed remains to be investigated. Interestingly, the NF1-associated myopathy might therefore be classified as satellite

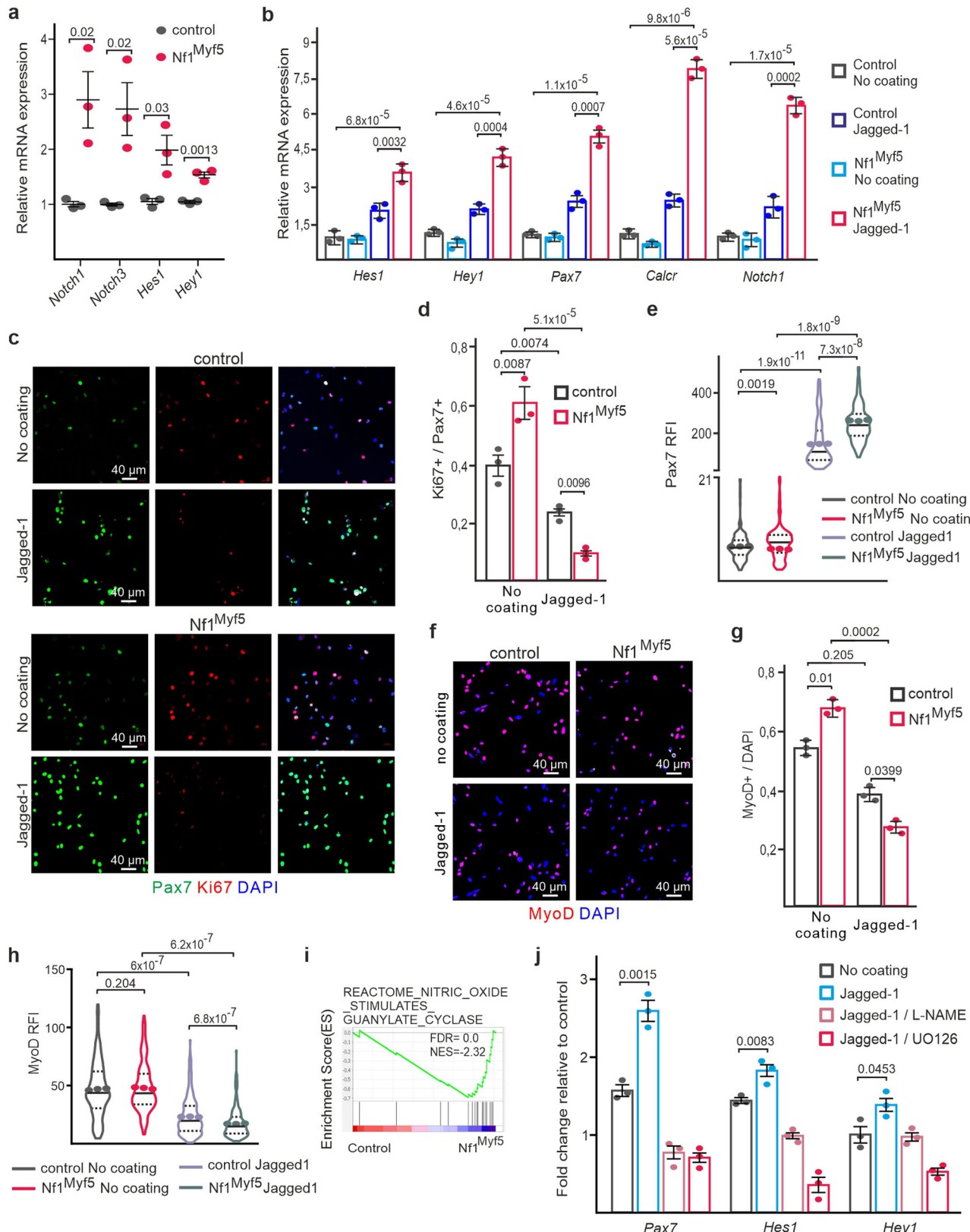

cell-opathy[84], as has also been suggested for several myopathies caused by mutations in Notch pathway components[85]. Early-life Notch pathway inhibition ameliorates muscle defects and the whole-body catabolic state of Nf1[Myf5] mice during the first three weeks of postnatal life. It will be interesting to see if this effect is persistent, or if treatment has to be repeated at later time points. Nevertheless, as the phenotype we observe in Nf1[Myf5] animals critically develops within the first weeks of life we propose that the efficacy of treatment targeting Nf1-dependent pathways towards improvement of muscular function may depend on a critical early postnatal time window.

**Fig. 6 | Increased Notch signaling drives Nf1$^{Myf5}$ MPs to quiescence. a** RT-qPCR of Notch pathway component and target genes in RNA extracted from freshly FACS-isolated control and Nf1$^{Myf5}$ p7 MPs ($n = 3$ animals per genotype; each dot represents the mean of three technical replicates from one biological replicate; $p$-values shown). **b** RT-qPCR for Notch targets on FACS-isolated control or Nf1$^{Myf5}$ p14 MPs cultured on Matrigel without coating or Jagged-1 coating for 48 h ($n = 3$ independent experiments from 3 animals per genotype; $p$-values shown). **c** FACS-isolated control or Nf1$^{Myf5}$ p14 MPs cultured on Matrigel without coating or Jagged-1 coating for 48 h stained for Pax7 (green) and Ki67 (red). **d** Ki67$^+$ cell quantification among Pax7$^+$ cells on image data as in (**c**) ($n = 3$ animals per genotype; $p$-values shown). **e** Anti-Pax7 relative fluorescence intensity (RFI) quantification on image data as in (**c**). Data range is shown as violin plot with median and interquartile range, means of biological replicates are shown as dots ($n = 3$ animals per genotype; $p$-values shown). **f** Immunolabeling for MyoD (red) on FACS-isolated control or Nf1$^{Myf5}$ p14

MPs cultured on Matrigel w/o coating or Jagged-1 coating for 48 h. **g** Quantification of MyoD$^+$ cells / total cells on image data as in (**f**) ($n = 3$ animals per genotype; $p$-values shown). **h** Quantification of anti-MyoD relative fluorescence intensity (RFI) on image data as in (**f**). Data range is shown as violin plot with median and interquartile range, means of biological replicates are shown as dots ($n = 3$ animals per genotype; $p$-values shown). **i** GSEA on RNA-Seq data from control and Nf1$^{Myf5}$ p7 MPs for "nitric oxide stimulates guanylate cyclase". **j** RT-qPCR for Notch targets *Pax7*, *Hes1* and *Hey1* on FACS-isolated control or Nf1$^{Myf5}$ p14 MPs. MPs were cultured on Matrigel without coating or with Jagged-1 coating for 48 h, with or without Mek inhibitor UO126 or pan-NOS inhibitor L-NAME. Bars show fold-changes of Nf1$^{Myf5}$ MPs relative to control MPs, control MPs were set as 1 ($n = 3$ independent experiments from 3 animals per genotype; $p$-values shown). Data are mean ± SEM; *P*-value calculated by two-sided unpaired *t*-test. Source data are provided as a Source Data file.

## Methods

### Animals and animal study approval

All animal procedures conducted within this study have been conducted in accordance with FELASA and ARRIVE guidelines and were approved by the responsible authority (Landesamt für Gesundheit und Soziales Berlin, LaGeSo) under license numbers ZH120, G0346/13, G0176/19 and G0270/18. Myf5$^{Cre86}$ and Acta1$^{Cre36}$ mice were obtained from Carmen Birchmeier (Max Delbrück Center for Molecular Medicine Berlin, Germany), Nf1$^{flox87}$ mice were obtained from The Jackson Laboratory (Nf1tm1Par/J), and Rosa26$^{mTmG37}$ mice were obtained from Andreas Kispert (Medizinische Hochschule Hannover, Germany). Both male and female mice were used and mixed in control and mutant groups. Mice were kept in an enclosed SPF facility with daily health monitoring, regular light/dark cycles, temperature of 22 °C, and 55% humidity. Mice were fed standard chow (ssniff V1124-000) ad-lib. Timed matings were set up, and mice were sacrificed by cervical dislocation, fetuses were sacrificed by decapitation.

### Primary cell isolation

Muscles from both fore- and hind limbs were minced and digested with 2 mg/ml collagenases A (Sigma-Aldrich, #11088793001) at 37 °C, for 1.5 h. Tissue was further homogenized with a syringe; cells were washed twice with FACS buffer (PBS supplemented with 2% FBS, 1 mM EDTA) and centrifuged with 300 g for 5 min at 4 °C. Tissue aggregates were removed using a 70 μm cell strainer (Miltenyi Biotec). Cells were re-suspended in FACS buffer and incubated with antibodies: CD45 (Thermo Fisher, # 17-0451-83; 1:100), CD31 (Thermo Fisher, # 17-0451-83; 1:100), Sca-1 (BioLegend, # 108126; 1:200), Ter119 (Thermo Fisher, # 17-5921-83; 1:100) and α7-Integrin (R and D Systems, # MAB3518; 1:200) on ice for 30 min. This was followed by centrifugation with 300 g for 5 min at 4 °C, and washing with FACS buffer. Cell sorting was performed sing FACS Aria II SORP (BD Biosciences). Propidium Iodide (eBioscience, # 00-6990-50; 1:1000) was used to evaluate cell viability. Myogenic progenitors were gated for α7-Integrin after eliminating all CD45, CD31, Sca-1, and Ter119 positive cells from all mononuclear cells. FACS-sorted cells were harvested into tubes containing FACS buffer (for RNA or gDNA isolation, and for cytospin) or proliferation medium for culture (DMEM with 20% FBS, 1mM L-Glutamine, 1× Penicillin/streptomycin).

FACS-isolated cells were centrifuged with 500 g for 5 min, re-suspended in proliferation medium and counted using an automated cell counter (Luna™). Desired cell numbers were seeded on 12 mm coverslips that were counted with Matrigel (10% Matrigel (Corning, # 356231) for 30 min). Cells were kept in proliferation medium, or immediately after adhesion subjected to differentiation conditions (DMEM with 2% HS, 1mM L-Glutamine, 1× Penicillin/streptomycin).

Primary myoblasts were isolated using pre-plating. After digestion (as above), the single-cell solution was put on plastic dishes for 2 h to separate adherent fibroblast. Supernatant was taken off and transferred to dishes coated with 0.1% gelatin (Sigma–Aldrich, # G1393).

Cells were cultured in proliferation medium for three days and detached using 0.25% trypsin-EDTA (Thermo Fisher, #11560626). Cells were again transferred to uncoated plastic dishes, left for 45 min, and supernatant was transferred to gelatin coated dishes.

### Single muscle fiber isolation and processing

EDL muscles from 15-week-old mice were isolated and digested in 2 ml of collagenase solution (0.2% collagenase type I in DMEM) at 37 °C in a water bath. During digestion the muscle was regularly checked to avoid over-digestion. Digestion was stopped by carefully transferring the muscle to a pre-warmed Petri dish. A large bore glass pipette was used to release single myofibres under a microscope. To re-equilibrate the medium during the procedure, every 10 min muscles were transferred to 37 °C and 5% CO$_2$ for 5 min. Fibers were fixed in pre-warmed 4% PFA/PBS for 5 min followed by washing in PBS for 10 min for 3 times. Fibers were incubated in 1% glycine in PBS to minimize PFA background staining. Fibers were permeabilised with 0.5% Triton X-100 in PBS for 10 min. For blocking, TSA blocking solution (Roche) was used at 4 °C overnight. Fibers were washed once in PBS for 5 min and incubated anti-Myosin, Skeletal, Fast (Sigma-Aldrich# M1570, 1:500) for 1 h at room temperature. Fibers were washed for 5 min for three times in PBS followed by incubation with secondary antibody and DAPI in staining solution (1× PBS, 0.03% horse serum, 0.003% BSA, 0.001% Triton X-100) for 1 h at room temperature. Fibers were washed in PBS for 5 min for three times. Then, fibers were individually transferred to a glass slide and mounted with Fluoromount-G. Fibers were visualized and photographed using an Axiovert 200 M (Zeiss) equipped with AxioVision 4.6 software (Zeiss). At least 30 fibers per sample were analyzed.

### Immunolabeling of cells

For cytospin analysis, FACS-sorted MPs were kept at 4 °C for 1 h followed by spinning at 50 g at 4 °C for 5 min. $5 \times 10^4$ cells were used per poly-L-lysine (1:100 dilution with bidest, Millipore, # A-003-E) coated 10 mm coverslip. MPs from cell culture were washed once with PBS to remove medium. Cells were fixed with 4% PFA (Merck, #104005) in PBS at room temperature for 10 min, permeabilized with PBX (0.3% Triton X-100) at room temperature for 10 min, and blocked with 5% BSA (Carl Roth, # 8076) in PBX at room temperature for 1 h. Primary antibodies were diluted in 5% BSA and cells were incubated at 4 °C overnight. After washing with PBX for 10 min three times, cells were incubated with secondary antibodies and DAPI (Invitrogen, #62248, 1:1000) diluted in PBX for 1 h at room temperature. Cells were washed 10 min for three times with PBX, and coverslips were mounted using Fluoromount-G (Southern Biotech, # 0100-01), and visualized using LSM700 confocal microscope (Zeiss) with ZEN imaging software (Zeiss). The primary antibodies and dilutions used were: Mouse anti-Pax7 (DSHB # Pax7; 1:25), Goat anti-Pax7 (provided by C. Birchmeier; 1:100), Rabbit anti-Ki67 (Abcam # ab16667; 1:500), Mouse anti-Ki67 (BD Biosciences #550609; 1:500), Rabbit anti-MyoD

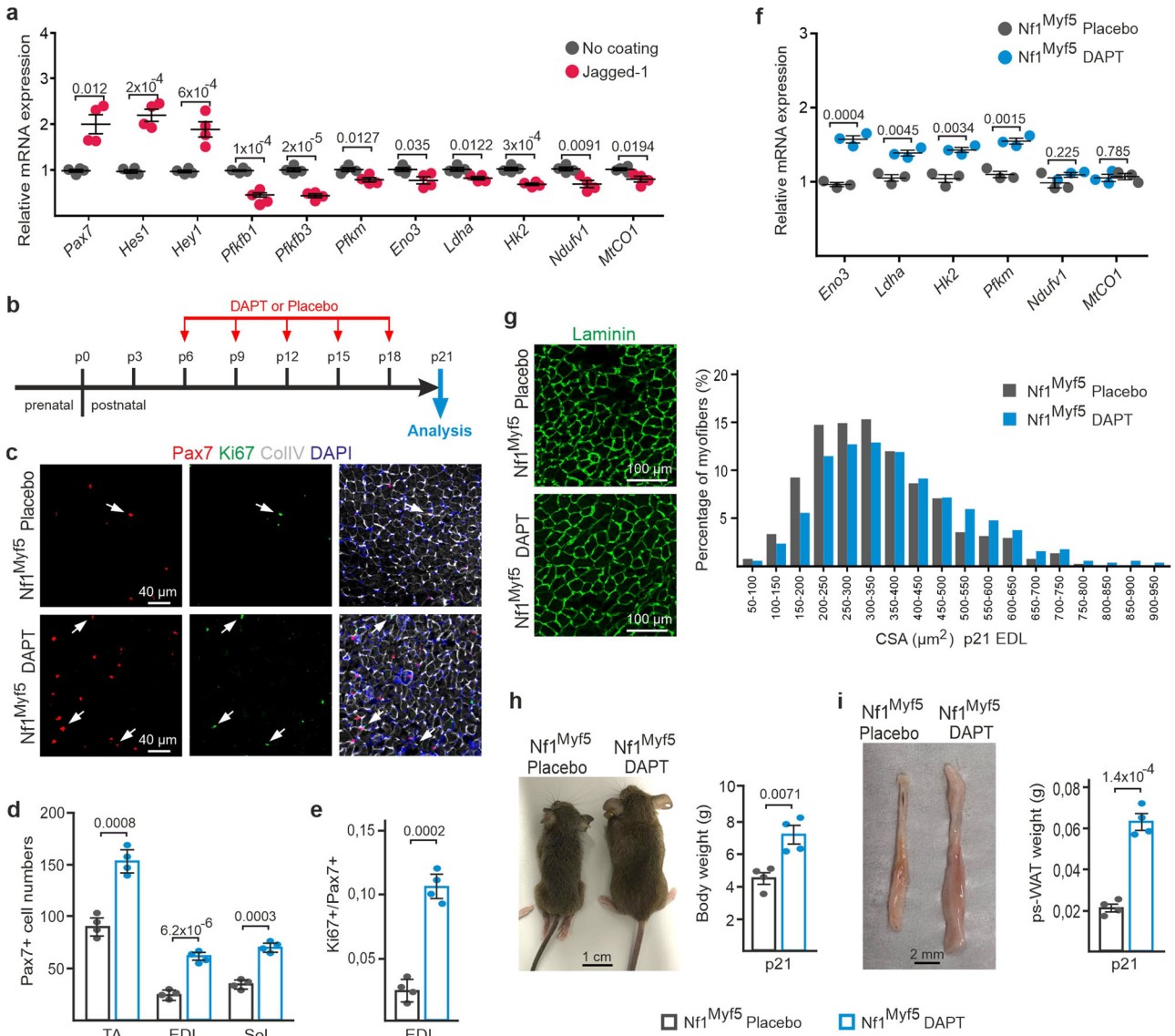

**Fig. 7 | Rescue of Pax7 cell depletion, cell cycle exit, and metabolic reprogramming by Notch pathway inhibition. a** RT-qPCR of selected glycolysis, TCA, and OXPHOS genes on FACS-isolated WT p14 MPs cultured on Matrigel without coating or Jagged-1 coating for 48 h. *Pax7*, *Hes1*, and *Hey1* tested as internal controls (*n* = 4 animals per condition; each dot represents the mean of three technical replicates from one biological sample; *p*-values shown). **b** Schematic depiction of DAPT treatment of Nf1^Myf5 animals. **c** Representative images of TA muscle sections of postnatal Nf1^Myf5 mice treated with placebo or DAPT, stained for Pax7 (red), Ki67 (green), collagen IV (gray), and DAPI (blue; nuclei). **d** p21 Pax7+ cell quantification in Nf1^Myf5 mice treated with placebo or DAPT (*n* = 4 animals per condition; *p*-values shown). **e** p21 Ki67+/Pax7+ cell quantification relative to Pax7+ cells in Nf1^Myf5 mice treated with placebo or DAPT (*n* = 4 animals per condition; *p*-value shown). **f** RT-

qPCR for glycolysis, TCA, and OXPHOS genes on muscle tissue from Nf1^Myf5 mice treated with placebo or DAPT (*n* = 3 animals per condition; each dot represents the mean of three technical replicates from one biological sample; *p*-values shown). **g** Representative images of Laminin (green) immunolabeling on sections of Nf1^Myf5 mice treated with placebo or DAPT shown left. Right: distribution of myofiber diameter in Nf1^Myf5 mice treated with placebo or DAPT (*n* = 4 animals per condition). **h** Body weight of Nf1^Myf5 mice treated with placebo or DAPT (*n* = 4 animals per condition; *p*-value shown). **i** Posterior subcutaneous white adipose tissue weight in Nf1^Myf5 mice treated with placebo or DAPT (*n* = 4 animals per condition; *p*-value shown). Data are mean ± SEM; *P*-value calculated by two-sided unpaired *t*-test. Source data are provided as a Source Data file.

(Cell Signaling Technology #13812; 1:300), Mouse anti-MyoD (BD Biosciences # 554130; 1:500), Mouse anti-MF20 (DSHB # MF20; 1:50), Anti-Myosin Skeletal, Fast (Sigma–Aldrich # M1570; 1:500), Rabbit anti- pErk1/2 (Cell Signaling Technology # 9101; 1:300), Rabbit anti-phosphor (s235/236)-S6 (Cell Signaling Technology # 4858; 1:300), Goat anti-Desmin (R&D Systems # AF3844; 1:500), Anti-trimethyl-Histone H3 (Lys27) (Millipore # 07-449; 1:500), anti-acetyl-Histone H4 (Lys16) (Millipore # 07-329; 1:500) (see also Supplementary Table 1). Relative fluorescence intensity (RFI) was measured with ImageJ software; at least 200 cells per biological sample and experiment were analyzed.

**Immunolabeling of tissue sections**

Mouse hind limbs were dissected and embedded with tragacanth (Sigma–Aldrich #G1128) on a cork plate followed by freezing in isopentane / dry ice for 10 s. Samples were stored at −80 °C. 10 μm sections (Microm HM355S) were used for all applications. For Pax7 staining, slides were fixed with pre-cooled methanol at −20 °C for 10 min, followed by antigen retrieval with antigen retrieval solution (2 mM EDTA) for 10 min at 95 °C in a water bath (Julabo). For all other antibodies, slides were fixed with PFA at room temperature for 10 min. Slides were blocked with blocking buffer (5% BSA in PBX) for 1 h at room temperature. Primary antibodies were diluted in blocking buffer

and slides were incubated overnight at 4 °C. For Pax7 staining, this was followed by at room temperature for 4 h. Primary antibody solution was removed, and slides were washed for 10 min for four times in PBX. Slides were incubated with secondary antibodies diluted in PBX for 1 h at room temperature, followed by washing with PBX for 10 min for four times. Slides were mounted with Fluoromount-G (Southern Biotech, # 0100-01) and visualized using LSM700 confocal microscope (Zeiss) with ZEN imaging software (Zeiss). The primary antibodies and dilutions used were: Goat anti-Collagen IV (Millipore AB769; 1:500), Rabbit anti-Laminin (Sigma–Aldrich L9393; 1:500), Mouse anti-Pax7 (DSHB Pax7; 1:10), Rabbit anti-Ki67 (Abcam ab16667; 1:400), anti-acetyl-Histone H4 (Lys16) (Millipore 07-329; 1:400), Mouse anti-β-Tubulin III (Sigma–Aldrich T8578; 1:100), Mouse anti-MyHC type 1 (DSHB BA-D5; 1:10), mouse anti-MyHC type 2 A (DSHB SC-71; 1:20), mouse anti-MyHC type 2B (DSHB BF-F3; 1:20), (see also Supplementary Table 1). TUNEL staining on tissue sections was performed using the DeadEnd™ kit (Promega) according to the manufacturer's instructions.

## Immunoblotting

For protein isolation, homogenization was performed with using TissueLyser (Qiagen) with RIPA buffer (50 mM Tris–Hcl, pH 8.0; 150 mM Nacl; 1% NP-40; 0.5% Sodium deoxycholate; 0.1% SDS). Protein concentration was measured using the Pierce BCA Protein Assay Kit (Thermo Fischer #23225). Total protein was separated with SDS-PAGE gels and transferred to PVDF membrane (GE Healthcare). Membrane were blocked with 5% BSA in TBST for 1 h at room temperature. Primary antibodies were diluted in blocking buffer, and membranes were incubated overnight at 4 °C room. After washing in PBST for 10 min for 3 times, HRP-conjugated secondary antibodies were applied for 1 h in PBS at room temperature. Images were acquired using a Fusion FX spectra gel documentation system (Vilber) with FUSION FX software. Primary antibodies used were: Rabbit anti-phosphor (Thr-389)-p70s6k (Cell Signaling Technology # 9205; 1:1000), Rabbit anti-p70s6k (Cell Signaling Technology # 9202; 1:1000). Blots images were analyzed and relative protein level was calculated using the gray value measurement tool in ImageJ.

## SeahorseXF metabolic flux analysis

Seahorse XF96 extracellular flux analyzer (Agilent) was used to measure the ECAR and OCR. The cartridge sensor was hydrated with 200 µl of calibration solution (Agilent) at 37 °C without $CO_2$ overnight. FACS-sorted MPs were plated on XF96 cell culture microplates coated with 10% Matrigel in warm assay medium (Agilent), the cell culture plate was centrifuged with 200 g for 5 min. and left 37 °C without $CO_2$ for 45 min. Measurements were performed using the glycolysis stress test kit (Agilent) according to the manufacturer's instructions. Seahorse Wave Desktop Software (Agilent) was used for data analysis.

## Jagged-1 ligand and inhibitor treatment

One day before the experiment, a 24-well plate was coated with 10% Matrigel at 37 °C for 30 min. Supernatant was removed and 100 µl of Jagged-1 ligand (5 ng/µl in PBS) was added (or only PBS for control plates), plates were left at room temperature overnight. Freshly FACS-sorted p14 control and Nf1$^{Myf5}$ muscle progenitors were cultured on Matrigel w/o coating or Jagged-1 coating for 48 h; $3 \times 10^5$ cells were used for each well. Cells were cultivated in proliferation medium without further treatment, or in the presence of Mek inhibitor UO126 (Promega # V1121; 10 µM) or pan-NOS inhibitor L-NAME (Sigma–Aldrich # N5751; 2 mM). Total RNA was isolated from each sample followed by RT-qPCR analysis for Notch target genes and selected glycolysis, TCA, and OXPHOS gene expression analysis.

## Notch signaling inhibitor injection

5 doses of γ-secretase inhibitor DAPT solution (Sigma–Aldrich #D5942; 30 mg/kg in 95% corn oil/5% ethanol) were applied by subcutaneous injection to four Nf1$^{Myf5}$ pups from postnatal day 6 to p21. Four Nf1$^{Myf5}$ pups injected with Placebo were used as controls. Animals were sacrificed at p21, and hind limb tissue was used for RNA extraction / RT-qPCR analysis, or cryosectioning.

## RT-qPCR

Total RNA was isolated using the RNeasy Micro Kit (Qiagen; tissue samples) or RNeasy Mini Kit (Qiagen; cells). cDNA was synthesized using 1 µg of mRNA using the SuperScript™ III Reverse Transcriptase (Invitrogen™ # 18080044) kit, Oligo(dT)20 primer (Invitrogen™ #18418020) and RNaseOUT™ Recombinant Ribonuclease Inhibitor (Thermo Fisher Scientific # 10777019) according to manufactures instructions. Quantitative RT-PCR analysis was performed using the ABI Prism HT 7900 real-time PCR detection system (Applied Biosystems) equipped with SDS software version 2.4 (Thermo Fisher Scientific) using GOTaq qPCR Master Mix (Promega) or SYBR Green qPCR Master Mix (Life Technologies). Actb was used to normalize the expression of each gene, double delta Ct (ΔΔCt) method was used to calculate the relative expression level. All primers used were purchased from Eurofins Scientific and listed in Supplementary Table 2. RT-qPCR was performed on at least 3 biological replicates (individual animals or independent cell culture assays) and were performed in triplicates for each sample.

## RNA-sequencing analysis

Two biological replicates for each genotype were analyzed. For each replicate, total RNA from $5 \times 10^5$ FACS-sorted p7 MPs pooled from two mice was isolated using the RNeasy Micro Kit (Qiagen). RNA quantity and quality were tested on a Qubit®Fluorometer (Invitrogen) and a Bioanalyzer2100 (Agilent). Sequencing libraries was prepared following Roche's "KAPA stranded mRNA Seq" library preparation protocol. 11 cycles of PCR were used for libraries amplification followed by sequencing with an Illumina HiSeq 4000 system in PE75bp mode. 45–72 million fragments were acquired in each sample. Mapping was performed using STAR 2.4.2a software with mouse genome (mm9). Read counts were generated with R Studio function Summarize Overlaps and normalized to RPKM based on the number of uniquely mapped reads. Differential expression analysis was performed with DESeq2 using default settings. Genes with an absolute fold change ≥ 2 and adjusted $p$-value ≤ 0.01 were used as significantly differentially expressed. GSEA analysis was performed with the entire gene list using GSEA software 4.0.1 desktop (Broad Institute). Curated KEGG gene set and all Gene ontology set from the Molecular Signature Database (MSigDB) were used for overrepresentation analysis. Web-based DAVID 6.8 was used for functional annotation of differentially expressed genes. Heatmaps were generated using R Studio 3.4.3.

## Evaluation of metabolic capacities by kinetic modeling

Metabolic capabilities of individual samples were evaluated using an established kinetic model of the energy metabolism[49] encompassing carbohydrate-, lipid- and amino acid metabolism including ATP generation and key electrophysiological processes at the inner mitochondrial membrane as described in[50]. The model describes the dynamic of metabolites and fluxes via ordinary differential equations taking into account the regulatory properties of the underlying metabolic enzymes, such as substrate affinities (Km-values), allosteric properties (Ki-values and Ka-values), and alterations in these parameters due to hormone-dependent phosphorylation[88]. Maximal enzyme activity was assumed to be proportional to protein expression. Individualized models for each sample were generated scaling the maximal enzyme activities (vmax) of each metabolic enzyme and transporter by the relationship

$$v_{\max}^{\text{animal}} = v_{\max}^{\text{control}} * E^{\text{sampel}} / E^{\text{control}} \tag{1}$$

where Econtrol is the average enzyme expr ession in all controls and Esample is the enzyme expression in the individual sample. Metabolic capacities were determined as described. MATLAB R2020b was used for running simulations and generating graphs of respective outputs.

## ChIP-sequencing (ChIPmentation)

Two biological replicates were used for H3K4me3, H3K27me3, H4K16ac. $1 \times 10^5$ FACS-sorted p7 MPs from one animal were used for each sample, and two samples were used for each genotype. Cells were fixed (150 µl of PBS/2%FBS/1% Formaldehyde) for 10 min at room temperature. Cells were sonicated using a Bioruptor (Diagenode) in sonication buffer (10 mM Tris−HCl pH 8.0, 0.25% SDS, 2 mM EDTA), 1× Roche complete Protease Inhibitor (Roche # 11697498001) with setting "high",15 × (30 s on / 30 s off). After short spin-down this was repeated. Sheared chromatin was cleared by centrifugation for 10 min at full speed at 4 °C, supernatant was transferred to a new 0.2 ml PCR stripe and 2 µl of antibody (Anti-acetyl-Histone H4 (Lys16), Millipore # 07-329; Anti-trimethyl-Histone H3 (Lys4) Millipore # 07-473; Anti-trimethyl-Histone H3 (Lys27) Millipore # 07-449) was added and incubates on a rotator at 4 °C overnight. Magnentic beads (Invitrogen #10003D or 10001D) were preparation by washing and blocking at 4 °C overnight with 0.1% BSA/RIPA buffer (see above). Chromatin was incubated with magnetic beads at 4 °C for 2 h. CHIPmentation reaction[89,90] was performed by incubating beads with tagmentation buffer (12.5 µl 2× TD buffer, 11.5 µl nuclease free water, 1 µl Tn5 enzyme (Illumina #FC-131-1024)) at 37 °C for 10 min. De-crosslinking was performed by incubating beads with ChIP elution buffer (1% SDS, 100 mM NaHCO3, 250 mM NaCl) with proteinase K (0.5 mg/mL) (Sigma−Aldrich # 70663) at 55 °C for 1 h, followed by incubation overnight at 65 °C. Samples were eluted using the MinElute kit (Qiagen #28004), followed by 16–18 cycles PCR amplification. AMPureXP beads (Beckman Coulter #A63881) were sued for library product size selection. Size selected library concentration was measured with the Qbit®Fluorometer (Invitrogen) using 1–2 ng/µl in a total volume of 13 µl. Library quality was assessed using the Bioanalyzer2100 (Agilent), showing a fragment peak around 280 bp. Primers used for amplification and barcoding of ChIPmentation libraries were purchased from Eurofins Scientific and are listed in Supplementary Table 3.

## ChIP-seq data analysis

Sequencing quality was evaluated by FastQC software. ChIP-seq reads (Illumina 2 × 75 base paired-end) were mapped to the mouse reference genome (mm9) using the default parameters of BWA MEM aligner using bwa v0.7.15[91]. BAM files were filtered using samtools rmdup[92]. Only unique mapped reads were used for the downstream analyses. MACS2[93] version 2.1.2[94] was used to call peaks at 5% FDR and annotated with overlapping and proximal genes in R with the ChIPpeakAnno library[95]. Coverage profiles represent Reads Per Million (RPM) values have been averaged over replicates, calculated using deeptools2 bamCoverage[96] and visualized in IGV/UCSC genome browser[97,98]. ChIP-seq read density profiles for region set summits were calculated using deeptools2 computeMatrix and plotted with plotHeatmap function. ±10 kb around the TSS for H3K27me3 and H4K16ac, ±1 kb around the TSS for H3K4me3 were plotted and averaged respectively. Differential binding analysis of CHIP-seq peak data was performed using the default DESeq2 analysis (FDR < 0.05; Padj<0.05)[99]. GSEA and GO analysis were performed as described for RNA-seq.

## MeDIP-sequencing

Genomic DNA from $1 \times 10^6$ FACS-sorted p7 MPs from one animal was isolated with the All Prep DNA / RNA / Protein Mini Kit (Qiagen) and used as one sample, and two samples were used for each genotype. Sample quality was measured using the Nanodrop 2000. The low-input MeDIP protocol[100] was used for library preparation. 80 ng of genomic DNA were fragmented to around 170 bp followed by End Pair

and A tailing with the End Pair mix (NEB Next Ultra DNA library preparation kit). After ligation of truncated TruSeq adapters with the NEB Next Ultra Ligation Module, samples were purified with the AmpureXP beads. Samples were mixed with IP buffer (MagMeDIP −kit; DIAGENODE) and denatured for 3 min at 95 °C. Denatured DNA was incubated with anti-5-meC-antibody and prewashed magnetic beads overnight at 4 °C on a rotator. After capturing, beads were washed, and methylated fragments were incubated for 5 min at 55 °C, and for 15 min at 100 °C in a proteinase K/elution buffer mix (1 µl of proteinase K in 100 µl DIB-buffer; Diagenode). Capture efficiency was determined by qPCR against spiked-in Lambda-DNA fragments in precapture and postcapture library samples. Libraries were amplified in a final PCR step using barcoded TruSeq primers (Index prime; 12 cycles) and size selected on a 2% agarose gel (230–320 bp, peak at 282–294 bp). Quality was assessed on an Agilent Bioanalyzer and library concentration was determined by Qbit and qPCR. Primers used for MeDIP Library preparation are shown in Supplementary Table 4.

## MeDIP-seq data analysis

Illumina 2 × 75 base paired-end reads where aligned to the mouse reference genome mm9 using bwa v0.7.15 and analyzed with QSEA Bioconductor package v.1.12.0[101]. CpG enrichment profiles were calibrated using blind calibration method. Differentially methylated regions (DMRs) where called at 250 base windows at an FDR of 10%, and annotated with promoter (TSS ± 500 bases), exonic, and gene body regions as well as model-based CpG islands (CGI)[102]. For GO term-based overrepresentation analysis, all DMRs overlapping a CGI where assigned to the next gene.

## Quantification and statistical analysis

Analysis of NGS experiments is outlined above. All other data quantification was based on at least three independent biological replicates (individual animals or independent assays), N-numbers are noted in the figure legends. From each sample, at least three technical replicates were performed. Error bars represent the standard error of the mean (SEM). Two-tailed Student's t-test with 95% confidence interval was used to evaluate the significance of differences between two groups. Quantification and alignments of NGS analysis for RNA-seq, ChIPmentation, and MeDIP are described in more detail in the "Methods" section above.

## Reporting summary

Further information on research design is available in the Nature Portfolio Reporting Summary linked to this article.

# Data availability

The sequencing raw data generated in this study have been deposited in the Gene Expression Omnibus (GEO) database under the Super Series accession number GSE159026. All other data generated in this study are provided in the Supplementary Information Source Data file. Source data are provided with this paper.

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

## Acknowledgements

We thank the animal facility of the Max Planck Institute for Molecular Genetics, Berlin for expert support; Carmen Birchmeier for providing mouse lines and anti-Pax7 antibody; Daniel M. Ibrahim for help with ChIP-Seq procedure; and Roswitha Merle (FU:stat team) for help with statistical analysis. Funding: Chinese Scholarship Council (CSC): XW. Sonnenfeld Stiftung Berlin: XW and SS. Deutsche Forschungsgemeinschaft DFG (SFB 1470 – A08 and project number 422215721): N.B.

## Author contributions

Conceptualization: XW, SS. Methodology: XW, ML, HW, SB, RG, NB. Main investigation: XW. Additional Investigation: AR, JF, SPK, GK, JOM, AM, NB. Writing – Original Draft: SS. Writing – Review & Editing: XW, SS. Funding Acquisition: XW, SS. Resources: BT, RG, SS. Supervision: SS

## Funding

## Competing interests

The authors declare no competing interests.
