## [Peer Review File · Nature Communications]

REVIEWER COMMENTS

Reviewer #1 (Remarks to the Author):

This manuscript by Wei et al is a follow up study of the previous paper describe the nf1 (myf5-cre) knockout phenotype in differentiated myofibers. The authors tried to explain the muscle defect in the mutant mice from the other side of the story, the stem cell deficiency. By using different methods, they want to show that during the early postnatal development the KO muscle stem cells get into quiescence earlier than WT ones. Therefore, the KO myofibers have fewer nuclei. The authors tried to attribute this early cell cycle exit to the abnormal notch signaling in the muscle stem cells.

Major points:

Figure 2D, in the western blot of whole muscle, it is surprising that the mutant (myf5-cre; nf1-flox) and control muscle only show subtle difference of nf1 protein expression. The authors provided a quantification in Figure 2E. It is still difficult to reconcile this modest difference in the protein levels to the major myofiber phenotype shift in the previous report.

Maybe the residue nf1 protein of the mutant mice is from other cells in the muscle tissue, i.e. fibroblast? Or it is from the non-myf5 lineage muscle cells?

Given that only marginal reduction of nf1 protein was observed in the myf5-cre:flox mice, it is also necessary to check whether the nf1 protein in the myofibers of acta-cre:flox mice is actually reduced or not.

The H3K27ac enrichment in quiescent muscle stem cells is still controversial. Some studies show that the H3K27me3 marked heterochromatin is the hallmark of the quiescence in muscle stem cells. In this study, the global H3K27me3 was dramatically reduced around TSS in nf1 KO cells (Fig 4C), but the authors only picked up two genes (pax7 and cdkn2a) to suggest the reduced H3K27me3 is associated with the impairment of proliferation. They should perform thorough functional annotation analysis of all the H3K27me3 target genes and whether H3K27me3 depletion on the TSS can increased the overall expression of the target genes.

Please perform the transcriptome PCA analysis of the WT and nf1 KO cells together with the in vivo quiescent muscle stem cells according to reference 40, 42. The FACS sorted cells are considered partially activated but not genuine quiescence.

Does the premature quiescence entry also affect the injury induced activation later on? Maybe this test is not relevant to the myofiber defects during development. But it would be interesting to know whether this early cell cycle exit population of muscle stem cells have the same activation capability?

Figure 5, the overlapped 130 down genes in both progenitor cells and differentiated cells were presented in some glycolysis pathways. The proportion of overlapped genes in each cell type seems quite low, which can't support the notion that the undifferentiated cells and differentiated cells share a metabolic alteration program. From the previous study, the KO myofibers showed a major metabolic disruption and the downregulated genes were all enriched in numerous metabolic pathways. But here only 130 out of 929 genes were shared with the undifferentiated cells. How about the rest of 799 genes, are they involved in metabolism of fibers? In addition, the authors claimed the KO fibers show enhanced oxidative metabolism in the previous paper but in this figure they show decreased oxidative activity in progenitors. If they share or inherit a common metabolic program, how to explain the opposite oxidative trend?

The Notch inhibition by dapt injection is not specific to stem cells. It is impossible to exclude the possibility that the metabolic changes in myofibers prevent the cell cycle exit of the associated muscle stem cell. More specific cell-autonomous methods should be performed.

The authors claim dapt treatment can rescue or partial rescue the defects of pax7 cells, myofiber size, body and fat weight. And in the figure 7 legend, they repeatedly say that dapt or placebo were given to control WT mice as well, but there is no control mice data at all? And also please show the real pictures for 7g, 7h, 7i.

Minor points:

Figure 1F, in figure legend, there is no top or bottom panels. Only left, middle and right panels.

Figure 1E, the Y axis, "% MyoD+ nuclei in MF20+ cells" is difficult to understand. The figure legend explains it very well but in the figure, it should change to another term.

Materials and Methods, "cells were washed cells twice with FACS buffer"

Page 9, line 4, please add one or two sentences explain why you should analyze erk1/2 in the mutant muscle.

The authors claimed the TSS H3K4me3 distributions are similar between KO and WT cells, but H3K27ac are very different, please provide statistical and significance analysis.

The dnmt3b expression is higher in mutant cells. Why the authors say dnmt3b has low expression?

Reviewer #2 (Remarks to the Author):

In this study, et al. report the discovery that mice with a satellite cell-specific neurofibromin deficiency have defective proliferation (overt quiescence) due to altered Notch signaling pathway activity. They used a combination of in vitro and in vivo methods, cell biology and genomics profiling, to arrive at their conclusions. They show that the mutant satellite cells have a downregulation of anabolic processes, decreased glycolysis.

The story is original, interesting and potentially impactful, but it is very complex and at times hard to follow, implicating myogenesis, metabolism, epigenomics (both histone and DNA methylation), the Ras/Mek/Erk pathway, the Notch pathway, the Sirt pathway.

Unfortunately, the study is riddled by many inaccuracies, misleading statements, omissions or poor data analysis. Several analyses and methods were insufficiently detailed, especially with regards to the genomics data analysis (RNA-seq, ChIP-seq, MeDIP-seq). Some conclusions are hastily drawn and not fully supported by the evidence. Most of these issues can be addressed by numerous yet straightforward revisions of the text and figures.

Specific comments

Fig. 1D: the authors note that freshly isolated satellite cells are less numerous to express MyoD and conclude of a "differentiation delay". Strictly speaking, this is more indicative of an "activation delay or impairment". To be able to characterize this as a delay, the authors must show that with longer time in culture, the mutant cells eventually reach the proportion of Pax7+/Myod+ cells attained in WT cells. The same comment applies to the myotube formation defect: the authors showed a single time point, so it is impossible to determine if there is a delay or a definitive impairment in differentiation.

The authors conclude at Fig 1E that "These results indicated cell cycle exit of Nf1Myf5 MPs that was not coupled to myoblast differentiation". This statement should be reformulated for clarity and to more accurately reflect the analyses that were performed: the cell cycle defect was evaluated *in vivo* in resting, non-injured muscle (where 70-80% of cells are not cycling) while the differentiation defect was evaluated *in vitro* after activation of stem cells by enzymatic digestion of muscle tissue (the authors obtain 40% ki67+ after isolation, compared to 20% in intact muscle, which agrees with reports that *in vitro* isolation of stem cells is sufficient to trigger their activation, see e.g. Machado et al, Cell Reports 2017, reference #42 in the authors' manuscript). It seems that a more accurate interpretation of the current results is "In *vivo*, Nf1Myf5 MPs are less proliferative than WT, and following activation *in vitro* they are also less prone to differentiate".

Fig. 1F: There is potential data misrepresentation. Upon close inspection, the high-magnification images look nothing like the low magnification areas that were marked by white squares. This applies to both the WT and mutant fiber images. They look like they were taken from completely different images. This is potentially very misleading. The authors are urged to clarify this and find matching images.

Related to experiments in Fig. 3, the authors wrote "freshly FACS-isolated p7 MPs, immediately fixed after isolation to preserve the *in vivo* condition". Considering the above comment about the stem cell isolation procedure being sufficient to cause their early activation and significant remodeling of their transcriptome, adding clarifications is very important. The authors seem to imply that the cells were fixed AFTER isolation by FACS and before RNA extraction. It is not clear: what was the point of fixation, if RNA was isolated immediately after fixation? The point of fixation is to do it *IN SITU*, as soon as the muscle is taken out of the mouse, so that the subsequent enzymatic digestion of tissue does not cause activation. The fixation of tissue/cells is not mentioned in the legend of Fig. 3 and is not described in the RNA-seq section of the methods. This is important to correct. RNA extraction from fixed samples and obtaining high quality sequencing libraries is not trivial. For how long were the tissue/cells fixed? What concentration of formaldehyde was used? Was formaldehyde quenched afterwards? How was the RNA extraction procedure adjusted to make it work with fixed samples? How were crosslinks reversed afterwards? The lack of clarity and the absence of all these details cast doubts on the rest of the RNA-seq data. The NCBI GEO entry GSE159026 for these samples also does not mention anything about fixation. In figure 3i, photos of representative stem cells are shown. They are circular, which according to Machado et al. is the shape of cells that were not fixed or fixed only after they were isolated.

The analyses leading to the heatmaps in Fig. 3E-H need to be much better described. What are the genes that compose these gene sets? They should be given as a supplementary table. What fraction of those is represented in the heatmaps as differentially expressed in mutant vs WT? How many more genes are NOT shown because not significantly different in the NF1 mutant? These results should also be shown as GSEA plots (with running enrichment score, etc), not only heatmaps. Also on these heatmaps the color legends are very hard to interpret because they are not symmetrical (e.g. the first one goes from -1.88 to 10.8, another goes from -5 to 10) and the choice of a three-colors palette makes little sense to show RPKM (estimated expression levels) values. The figure legend and the methods offer no help to guide the

reader in properly interpreting these plots. Instead of showing RPKM, the authors should show log(fold change) or Z-score or median-centered RPKM values, which is customary with this sort of analyses. In this case, a three-colors heatmap color palette will make sense (below zero, zero, above zero). If they want, they can show also the average RPKM values as continuous variable row annotation.

The authors compare their dataset to that of Ryall et al. (ref 38). They should explain briefly what is that dataset and why it was chosen over the others presented in Fig. 3. Also they should explain in more details how the comparison (in Fig. S3E) was made. GSEA, using custom gene sets created from the Ryall et al data, could also be performed here.

In Fig. 4, the authors point that "condensed chromosome" was a significant gene set in their GSEA and thus conclude that this indicates chromatin alterations in the Nf1 mutants. This is over-simplistic. First, this is not indicative but "suggestive". Second, "condensed chromosome" contains 303 mouse genes (it is quite vast), including many falling under children terms associated with mitosis (e.g. the cohesin complex). So enrichment of "condensed chromosome" genes could simply be a reflection of a defective cell cycle: the mutants cycle less and thus express less genes associated with the M phase of the cell cycle.

Fig. 4B.: There are a few problems with this figure. It clearly shows two curves that do not overlap at the center, yet the authors wrote "...showed apparently unchanged H3K4me3 levels between controls and Nf1Myf5 MPs". Perhaps they were misled by the figure showing the average signal over an overtly broad 20 kb surrounding gene TSS. If they had chosen to focus on the TSS region (which is where H3K4me3 concentrates), by showing only ~1 kb on each side of it, the difference would be more obvious. Secondly, this is presumably the average signal for all genes in the genome (the authors forgot to give any detail anywhere). So there are possibly some genes showing higher signal and others showing lower signal. Showing not only the mean but also the standard deviation would be very informative.

The authors wrote "acquisition of a quiescent state in Nf1Myf5 MPs is concomitant with reduced repressive H3K27me3 modification at quiescence-related genes." There are two problems with this statement. First, it is based on an analysis of all genes on the genome (Fig. 4c) and only two specific genes implicated in quiescence (Fig. 4d and S4b), which do not justify the level of generalization the authors are conveying. Second, the statement implies that there are no equivalent changes at genes with the opposite function (i.e. promoting proliferation). To support their statement, the authors should show the plot in Fig. 4C but broken down to show separately the situation at cell cycle promoting genes, and at cell cycle arrest genes.

The ChIP-seq analysis method also does not explain how a quantitative analysis of H3K4me3 and H3K27me3 was made, e.g. there is no mention of internal standard spike-in chromatin to ensure that the WT and mutant datasets could be directly compared in a reliable manner. No control loci with invariant

levels of these marks are shown either. The authors only mention "normalized ChIP seq profiles" with no procedural details.

In Fig. S4, the RPKM values calculated from Ryall et al. (GSE64379) and from this study cannot be directly compared, there are too many different variables (for instances the library preparation kits were different, Ryall was single-end (Ryall) vs paired end (this study)). What may be compared is the fold-change from each study (quiescent vs activated, for Ryall et al.)

For MeDIP-seq analysis (Fig. 4G), the authors need to better explain how the results were obtained and how to interpret them (what are odds ratio and the ratios shown (100/64582)?).

The authors conclude from their epigenomic profiling that "epigenetic alterations observed in Nf1Myf5 MPs are consistent with a shift toward quiescence". However, although the validation work is nicely done, their analysis was quite superficial. A more accurate sentence would be "epigenetic alterations in Nf1Myf5 MPs, at the genes we subjected to validation, are consistent with a shift toward quiescence". It would also be important to better explain the rationale for selecting the specific metabolism genes that went through this validation: what links them to quiescence exactly?

The legend to Figure 5 needs to be more clear on how genes make it to be represented on the various heatmaps: are they all genes in these GO categories? Are they all significant differentially expressed genes? Are they the "core enrichment genes" from the GSEA analysis?

On page 17, the authors wrote "As Nf1Acta1 mice show no muscle defect, this suggests that perturbed muscle fiber metabolism in Nf1Myf5 animals 30 can be traced back to defects in juvenile MP".

In reality, the characterization of the Nf1Acta1 animals (Fig 2A, B, C) was superficial, focusing only on gross animal aspect and myofiber sizes. These animals (or their myofibers) were not analyzed in terms of their metabolism, they were not subjected to RNA-seq analysis, etc, so the statement has to either be removed or be backed with additional data for a direct comparison of the two strains (with Acta1-Cre and Myf5-Cre).

Fig. 6J: the authors tested the effect of blocking the NOS pathway in Jagged-treated cells. However, they omitted to also treat similarly the No-Jagged control cells. Their conclusion that "pan-NOS inhibitor L-NAME cancelled the hyper-responsiveness to Jagged-1" is thus not fully supported, since in principle L-NAME could also act independently of Jagged-1. The statement should be revised, or new data showing that L-NAME has no effect in the absence of Jagged-1 stimulation should be presented.

page 23: The following statement is not fully supported by the data: "This indicated that aberrant Notch pathway activity was not abrogated but effectively blunted by our DAPT regime in Nf1Myf5 mice." Making the call between abrogation and blunting (taken to mean "partial decrease") would require more evidence than just qRT-PCR (e.g. showing that cells have a decreased response to Jagged-1). For one, it could be that the effect at the level of protein abundance is much more pronounced than at the steady state mRNA level.

To make a link between the altered metabolism (reduced glycolysis) and the Notch pathway, the authors invoke the overlap between Sirt1 target genes (identified by Ryall et al.) and profiles of H4K16 acetylation (a mark erased by Sirt1). They conclude in the discussion "H4K16ac decoration is highly reduced in Nf1Myf5 juvenile MPs. Myogenic genes affected in Nf1Myf5 MPs overlap genes identified as Sirt1 targets in MuSCs, indicating Sirt1-mediated metabolic reprogramming of Nf1Myf5 juvenile MPs, preventing myogenic differentiation. " However, the verb "indicate" feels too strong, considering the evidence presented: the authors do not offer any convincing validation of this hypothesis that Sirt1 is really involved in metabolic reprogramming in the Nf1Myf5 mice. Supporting evidence (more than correlative observations) would need to be provided, or the statement should be revised.

Other notes:

page 3: "and their transmission to quiescence." Probably the authors mean "transition"

page 5: "Proliferation rates of Nf1Myf5 Pax7+ cells assessed by Ki67 immunolabeling". Staining with Ki67 does not give the "rate of proliferation" but rather the proportion of cells actively proliferating. Growth curves would be needed to assess the proliferation rate.

Fig 1c: to be informative, the IF image should show the two color channels separately, as in Fig. 1B. The Pax7 intensity measurement should be a separate figure panel (with its own letter label). The legend does not explain what the horizontal black bars represent (the mean? the median?). Those data should be shown as a boxplot (or better, as a violin plot) to better represent the distribution of values.

Fig. 1E: the y-axis label indicates "% myod+ nuclei in MF20+ cells", and shows a range from 0 to 0.6. These are very low percentages. Did the authors forget to multiply by 100? Or perhaps they meant "PROPORTION of myod+ nuclei..."?

Acta1-Cre: To avoid any misunderstanding, the authors should specify in the relevant methods section that this Cre driver is a transgenic mouse line expressing Cre recombinase under the control of the human alpha-skeletal actin promoter. Maybe the Jax stock number can be given too.

Fig 2d: The authors wrote that Nf1 is "barely detectable" in muscle lysates, but this feels like an overstatement. The protein is perfectly detectable and simply less expressed compared to myoblasts.

The main text and methods should mention how many biological replicates were used for the RNA-seq experiment (supp figure 3 indicates 2 of each genotype).

Figure 3: the panel letters in the figure and in the legend do not all match (e.g. panel d is GSEA in the figure but fluorescence signal in the legend).

Fig 3.A: the volcano plot show the $-\log_{10}$ of pvalue, but it is the pvalue after multiple hypothesis testing that must be shown instead.

Fig. 4G: the x-axis indicates "CpG insland", I presume it should be "island"

page 17: "and decreased H4K14ac decoration", it should be H4K16ac

The legend to Fig 6A should make it clear that the cells being analyzed are (e.g. "RNA extracted from MPs immediately FACS"). Otherwise the reader may get confused with panel 6B (MPs put in culture for 48h)

Fig 6C: the multi-channel IF needs to be split into individual channels, because the green color (Pax7 signal) is strong strong, it hides all other colors.

page 21: the authors omitted to explain what DAPT is, and what is the rationale for using it in their experiment.

page 23: "myonucelar accrual"

Reviewer #3 (Remarks to the Author):

The investigators use conditional Nf1 knockout mice to demonstrate that Nf1 is not required in muscle fibers, but rather postnatal myogenic progenitors. They further demonstrate that this requirement operates at the level of cell cycle exit, quiescence, and differentiation, which is dependent on Mek/Erk/Nos signaling. This Mek-driven effect sensitizes Nf1-deficient MPs to Notch signaling. Taken together, these observations extend what we know about neurofibromin regulation of metabolism.

Since patients with NF1 are heterozygous for a loss of function NF1 allele, it would be important to analyze MPs heterozygous for a loss of function Nf1 allele in order to be relevant to the human condition.

Given the reduction in Myf5 levels in these mice, were Myf5-Cre mice identical to wild-type mice in terms of MP gene expression and function?

Figure 1. Was there any change in MP apoptosis or premature senescence?

Figure 2. Size markers should be included for all Western blots. What is the molecular size of the neurofibromin band shown?

Figure 4. Is N-Ras the predominant hyperactivated Ras in Nf1-deficient MPs? Some of the changes in RNA expression, while statistically different, look minimally different (<10-20% relative to controls). The results in panel j should be normalized to total p70S6K, rather than actin, to determine the relative reduction in activated p70S6K in Nf1-deficient MPs. Since neurofibromin has previously been shown to result in increased mTOR signaling, which should increase p70S6K phosphorylation, what do the authors think is responsible for this finding?

REPLIES TO REVIEWER COMMENTS

Reviewer #1 (Remarks to the Author):

This manuscript by Wei et al is a follow up study of the previous paper describe the *nf1* (*myf5*-cre) knockout phenotype in differentiated myofibers. The authors tried to explain the muscle defect in the mutant mice from the other side of the story, the stem cell deficiency. By using different methods, they want to show that during the early postnatal development the KO muscle stem cells get into quiescence earlier than WT ones. Therefore, the KO myofibers have fewer nuclei. The authors tried to attribute this early cell cycle exit to the abnormal notch signaling in the muscle stem cells.

Major points:

Figure 2D, in the western blot of whole muscle, it is surprising that the mutant (*myf5*-cre; *nf1*-flox) and control muscle only show subtle difference of *nf1* protein expression. The authors provided a quantification in Figure 2E. It is still difficult to reconcile this modest difference in the protein levels to the major myofiber phenotype shift in the previous report. Maybe the residue *nf1* protein of the mutant mice is from other cells in the muscle tissue, i.e. fibroblast? Or it is from the non-*myf5* lineage muscle cells?

Thank you very much for this comment. Indeed, in Fig. 2d (now supplementary Fig. 2g) shows residual *Nf1* protein expression in *Nf1Myf5* homozygous KO muscle, thus suggesting only a moderate downregulation. We have

- analyzed *Nf1* mRNA expression during myogenic differentiation of primary myoblasts (Fig. 2d) reconfirming the downregulation of *Nf1*.
- We have also analyzed *Nf1* mRNA levels in whole muscle tissue versus isolated primary myoblasts and fibroblasts (Fig. 2e), showing exactly what the reviewer suggested; *Nf1* is mainly expressed in fibroblasts in muscle tissue, thus explaining the residual expression in *Nf1Myf5* mice, where fibroblasts are unaffected.

Given that only marginal reduction of *nf1* protein was observed in the *myf5*-cre:flox mice, it is also necessary to check whether the *nf1* protein in the myofibers of *acta*-cre:flox mice is actually reduced or not.

In the line of the above comment and our data we propose that such analysis would very likely also only show a marginal overall reduction, as fibroblasts, and in this case also MuSCs, are unaffected. Thus, we think that the value of this analysis is low, we hope the reviewer agrees. We also note that we have considerable difficulties with a new batch of the *NF1* antibody we had previously used, precluding this analysis at the moment.

The H3K27ac enrichment in quiescent muscle stem cells is still controversial. Some studies show that the H3K27me3 marked heterochromatin is the hallmark of the quiescence in muscle stem cells. In this study, the global H3K27me3 was dramatically reduced around TSS in *nf1* KO cells (Fig 4C), but the authors only picked up two genes (*pax7* and *cdkn2a*) to suggest the reduced H3K27me3 is associated with the impairment of proliferation. They should perform thorough functional annotation analysis of

all the H3K27me3 target genes and whether H3K27me3 depletion on the TSS can increased the overall expression of the target genes.

Thank you very much for pointing this out, we apologize for the shortcoming. We have now improved this analysis by -performing DiffBind analysis to identify genes with significantly reduced H3K27me3 in Nf1Myf5 MPs; we provide this gene list as Supplementary Data 3, and performed GO analysis (Fig. 4d). We have furthermore intersected these genes with genes upregulated in Nf1Myf5 MPs (Fig. 4e; gene list in Supplementary Data 4) and performed GO analysis on the overlapping genes (Supplementary Fig. 4a).

This comment was particularly helpful: Pax7, which we picked to illustrate reduced apparent H3K27me3 was indeed contained in this intersected gene list (Supplementary Data 4; Fig. 4e).

Since the overlap between H3K27me3 genes and upregulated genes was, however, low, we added the following sentence (p14 lines 279-282): “Intersecting genes with reduced H3K27me3 levels with genes upregulated in Nf1Myf5 p7 MPs showed only a low overlap of 248 genes (Fig. 4e; gene list in Supplementary data 4), suggesting that reduced H3K27me3 alone cannot explain gene deregulation in Nf1Myf5 MPs.”

Please perform the transcriptome PCA analysis of the WT and nf1 KO cells together with the in vivo quiescent muscle stem cells according to reference 40, 42. The FACS sorted cells are considered partially activated but not genuine quiescence.

We had in fact been thinking about this comparison before, but decided against it for two reasons: First, we did not perform in-situ fixation, so our dataset is not equal to the dataset of Machado et al. (doi: 10.1016/j.celrep.2017.10.080) We used standard FACS isolation, thus our dataset holds the same preactivation bias as e.g. the dataset of Ryall et al (doi: 10.1016/j.stem.2014.12.004), we thus think this is the appropriate comparison. Second, our dataset comes from 7 days postnatal. In this situation, MPs comprise a variety of states and only a fraction of them has undergone quiescence, unlike in adult muscle, where >95% or more can be considered truly quiescent. For both reasons our mutant dataset, where we observe a transcriptional shift towards quiescence transcriptionally at p7, but not complete quiescence (no cell cycle exit yet; Fig. 1a), will not match the truly quiescent dataset. We have nevertheless performed the PCA analysis and for now display the result here for the reviewer.

Interestingly, while as predicted above, our datasets group apart from the adult truly quiescent and 3h preactivated datasets, the Nf1Myf5 dataset at least shows a shift towards the truly quiescent set. If the reviewer thinks it is worth showing these data we are happy to provide them in supplement.

We have furthermore used normalized read counts from our analysis and compared our control and Nf1Myf5 cells to normalized read counts of the truly quiescent cells (termed “Quiescent” in the images) of the Machado (doi: 10.1016/j.celrep.2017.10.080) and Van Velthoven (doi: 10.1016/j.celrep.2017.10.037) datasets, also confirming a quiescence shift. This could be added as well, if the reviewer sees fit.

Does the premature quiescence entry also affect the injury induced activation later on? Maybe this test is not relevant to the myofiber defects during development. But it would be interesting to know whether this early cell cycle exit population of muscle stem cells have the same activation capability?

We thank the reviewer for the interesting suggestion, but we think this is out of scope of the current manuscript. We will address this in the future.

Figure 5, the overlapped 130 down genes in both progenitor cells and differentiated cells were presented in some glycolysis pathways. The proportion of overlapped genes in each cell type seems quite low, which can't support the notion that the undifferentiated cells and differentiated cells share a metabolic alteration program. From the previous study, the KO myofibers showed a major metabolic disruption and the downregulated genes were all enriched in numerous metabolic pathways. But here only 130 out of 929 genes were shared with the undifferentiated cells. How about the rest of 799 genes, are they involved in metabolism of fibers? In addition, the authors claimed the KO fibers show enhanced oxidative metabolism in the previous paper but in this figure they show decreased oxidative activity in progenitors. If they share or inherit a common metabolic program, how to explain the opposite oxidative trend?

We are not sure we understand the comment correctly. Firstly, we compare the transcriptomes of 7 day postnatal myogenic progenitors and p21 whole muscle tissue, thus only a partial overlap in gene expression profiles can be expected between both in the first place. The image below shows a PCA analysis comparing only control p7 MPs and control p21 muscle tissue, demonstrating that expectedly their transcriptomes are distinct:

What we show is, amongst deregulated genes in Nf1Myf5 animals, a small proportion is shared between p7 MPs and p21 muscle tissue. As mentioned above, we compare different cell types, so we find even this overlap in common deregulation striking and worth analyzing. However, this overlap does not simply consist of a common deregulation of myogenic genes, as would be expected from the analyses shown in Fig. 3. What we show is a clear overlap in a very specific signature commonly downregulated between Nf1Myf5 p7 MPs and Nf1Myf5 p21 muscle tissue: genes belonging to glycolysis and biosynthetic pathways. What we state is thus, that this deregulation is conserved (while other parts are not). We think this conclusion is clear and justified. We have now expanded the metabolic analysis of Nf1Myf5 p7 MPs and p21 muscle tissue by performing in silico metabolic flux modeling, which confirms our present analysis and also our previous data (Wei et al. 2020 JCSM, doi: 10.1002/jcsm.12632), now shown in Supplementary Fig. 5. This analysis also confirms that the common metabolic denominator between Nf1-deficient MPs and muscle tissue is impaired glucose metabolism.

Indeed, genes belonging to oxidative metabolism are conversely regulated between p7 MPs and p21 muscle tissue. In silico metabolic flux modeling confirmed this (Supplementary Fig. 5). Furthermore, this was confirmed experimentally: p7 MPs showed no significant alteration in OCR, but severe reduction of ECAR (Seahorse metabolic flux assay, Fig. 5e, f); conversely, p21 muscle fibers showed increased oxygen consumption as evidenced by Oroboros assay (Wei et al. 2020 JCSM, doi: 10.1002/jcsm.12632).

Thus, this metabolic signature apparently is not conserved between MPs and muscle tissue. We interpret this as a consequence of the two cell types analyzed: at p7 we analyze progenitors; here, Nf1Myf5 MPs shift to a quiescent state with low energy demand, hence no need for enforced oxidative metabolism even despite strongly impaired glycolysis. In p21 muscle the situation is opposite; this tissue has high energy demand, which in fast-twitching muscle is mainly met by glucose metabolism. Since our RNA-Seq data and metabolic flux modeling based on this suggest conserved impairment of glycolysis, muscle fibers have no other choice than to rely on oxidative metabolism (mainly consuming fatty acids, but still showing an energy deficit as we have shown before, Wei et al. 2020 JCSM). In silico kinetic metabolic flux analysis (Supplementary Fig. 5) supports this idea; Nf1-deficient p7 MPs are predicted to have decreased maximal capacity for ATP production and oxygen consumption, however under homeostatic conditions apparently these capacities are sufficient to meet the needs of the cells as suggested by OCR analysis. We have included this thought in the text during the description of the Seahorse data, p18 lines 354-357: *"In contrast, Nf1Myf5 p7 MPs showed only a moderate decrease in the basal oxygen consumption rate (OCR) below statistical significance in (Fig. 5f), indicating that oxidative phosphorylation capacity is still sufficient to maintain resting energy demand in MPs."*

We thank the reviewer for the suggestion to also analyze the non-shared 799 genes; this analysis is now shown as Fig. S5d; in the text we added (p18/19 lines 368-370): “GO analysis of 799 genes downregulated in p21 Nf1Myf5 muscle, but not in p7 MPs, did not yield any metabolism-related terms (Supplementary Fig. 5d), but rather terms as “Z-disc” possibly reflecting fiber atrophy.”

Altogether, we are therefore convinced that the data we present support a scenario where glycolytic genes (via a mechanism yet to be revealed) are continuously repressed, while oxidative metabolic genes are not under such permanent inhibition. We have added the following sentence to the discussion to clearly state that at present we do not understand the basis for this finding (p32 lines 641/642): “Why in our model specifically glycolytic genes are continuously repressed remains to be investigated.”

The elucidation of the exact mechanism behind this permanent inhibition will certainly be very interesting, but clearly out of the scope of the present manuscript.

The Notch inhibition by dapt injection is not specific to stem cells. It is impossible to exclude the possibility that the metabolic changes in myofibers prevent the cell cycle exit of the associated muscle stem cell. More specific cell-autonomous methods should be performed.

We agree with the reviewer that this systemic delivery does not prove a cell-autonomous function. However, this experiment is accompanied by a large set of in vitro data that clearly demonstrate a cell-autonomous effect of Notch signaling on MPs, which are in line with the in vivo data. The purpose of this experiment was not primarily to demonstrate a cell-autonomous role for Notch signaling in the pathogenesis of the myopathy, but to rather demonstrate that the detrimental activity of the Notch pathway can be targeted in vivo. We have taken up this point in the discussion and added a sentence on p29/30 lines 579-581: “Although in this scenario we cannot exclude that Notch inhibition affects other cells apart from MPs, the outcome is in line with the cell-autonomous effects observed in vitro.”.

The authors claim dapt treatment can rescue or partial rescue the defects of pax7 cells, myofiber size, body and fat weight. And in the figure 7 legend, they repeatedly say that dapt or placebo were given to control WT mice as well, but there is no control mice data at all? And also please show the real pictures for 7g, 7h, 7i.

We apologize for this error; we only had ethical approval to treat Nf1 mutants with either placebo or DAPT, no wild type data are shown. The figure legend has been adapted.

We have added representative images for Fig. 7g, h, i.

Minor points:

Figure 1F, in figure legend, there is no top or bottom panels. Only left, middle and right panels.

Thank you; corrected

Figure 1E, the Y axis, “% MyoD+ nuclei in MF20+ cells” is difficult to understand. The figure legend explains it very well but in the figure, it should change to another term.

We changed this to “MyoD+ nuclei in Mf20+ myotubes (%)”.

Materials and Methods, “cells were washed cells twice with FACS buffer”
corrected

Page 9, line 4, please add one or two sentences explain why you should analyze erk1/2 in the mutant muscle.

We have removed this piece of data, as it deviates from the main message.

The authors claimed the TSS H3K4me3 distributions are similar between KO and WT cells, but H3K27ac are very different, please provide statistical and significance analysis.

The plots shown in Fig. 4a, b show averaged normalized coverage across all genes, and as such are descriptive. At least to our understanding it is not customary in the literature to use statistics on these depictions. We note that, as elaborated above, we have now performed a more in-depth analysis of H3K27me3 ChIP-Seq data. We have attempted to be as careful as possible with assumptions based on ChIP-Seq data; we note, however, that the global downregulation of H3K27me3 was independently confirmed by immunolabeling including statistical analysis. We have further clearly noted the limited quantitative information of our ChIP-Seq data in the discussion (p29 lines 556-559 “We, however, note that our analysis did not employ e.g. chromatin spike-in. While we confirmed global deregulation of H3K27me3 and H4K16ac with alternative methods, quantitative assumptions at individual loci for all marks we analyzed should be taken with caution.”).

The dnmt3b expression is higher in mutant cells. Why the authors say dnmt3b has low expression?

Low expression of *Dnmt3b* means that the mRNAs showed an overall low abundance in p7 MPs based on the RPKM values obtained from RNA-Seq (average RPKM for controls 0,58, for Nf1Myf5 2,83; see Supplementary Table 1). We have added made this clearer: o16 line312: “Dnmt3b had low expression levels based on RPKM values”.

Reviewer #2 (Remarks to the Author):

In this study, et al. report the discovery that mice with a satellite cell-specific neurofibromin deficiency have defective proliferation (overt quiescence) due to altered Notch signaling pathway activity. They used a combination of in vitro and in vivo methods, cell biology and genomics profiling, to arrive at their conclusions. They show that the mutant satellite cells have a downregulation of anabolic processes, decreased glycolysis.

The story is original, interesting and potentially impactful, but it is very complex and at times hard to follow, implicating myogenesis, metabolism, epigenomics (both histone and DNA methylation), the Ras/Mek/Erk pathway, the Notch pathway, the Sirt pathway.

Unfortunately, the study is riddled by many inaccuracies, misleading statements, omissions or poor data analysis. Several analyses and methods were insufficiently detailed, especially with regards to the genomics data analysis (RNA-seq, ChIP-seq, MeDIP-seq). Some conclusions are hastily drawn and not fully supported by the evidence. Most of these issues can be addressed by numerous yet straightforward revisions of the text and figures.

We apologize for these shortcomings and hope that our revised version resolves these issues.

Specific comments

Fig. 1D: the authors note that freshly isolated satellite cells are less numerous to express MyoD and conclude of a "differentiation delay". Strictly speaking, this is more indicative of an "activation delay or impairment". To be able to characterize this as a delay, the authors must show that with longer time in culture, the mutant cells eventually reach the proportion of Pax7+/Myod+ cells attained in WT cells. The same comment applies to the myotube formation defect: the authors showed a single time point, so it is impossible to determine if there is a delay or a definitive impairment in differentiation.

Thank you for pointing this out, we agree. We have rephrased this part on p6 lines 117-120: "Cytospun p14 MPs showed a relative decrease in MyoD+/Pax7+ cell numbers (Fig. 1d). Freshly isolated Nf1Myf5 p14 MPs plated in high density and immediately subjected to differentiation conditions showed a strong decrease in myotube formation compared to control MPs (Fig. 1e)."

The authors conclude at Fig 1E that "These results indicated cell cycle exit of Nf1Myf5 MPs that was not coupled to myoblast differentiation". This statement should be reformulated for clarity and to more accurately reflect the analyses that were performed: the cell cycle defect was evaluated in vivo in resting, non-injured muscle (where 70-80% of cells are not cycling) while the differentiation defect was evaluated in vitro after activation of stem cells by enzymatic digestion of muscle tissue (the authors obtain 40% ki67+ after isolation, compared to 20% in intact muscle, which agrees with reports that in vitro isolation of stem cells is sufficient to trigger their activation, see e.g. Machado et al, Cell Reports 2017, reference #42 in the authors' manuscript). It seems that a more accurate interpretation of the current results is "In vivo, Nf1Myf5 MPs are less proliferative than WT, and following activation in vitro they are also less prone to differentiate".

Thank you for the comment, we have deleted the above-mentioned statement and have taken over the reviewer's suggestion, see above.

Fig. 1F: There is potential data misrepresentation. Upon close inspection, the high-magnification images look nothing like the low magnification areas that were marked by white squares. This applies to both the WT and mutant fiber images. They look like they were taken from completely different images. This is potentially very misleading. The authors are urged to clarify this and find matching images.

Thank you very much for pointing out this error, we seriously apologize! We have corrected this and replaced the images.

Related to experiments in Fig. 3, the authors wrote "freshly FACS-isolated p7 MPs, immediately fixed after isolation to preserve the in vivo condition". Considering the above comment about the stem cell isolation procedure being sufficient to cause their early activation and significant remodeling of their transcriptome, adding clarifications is very important. The authors seem to imply that the cells were fixed AFTER isolation by FACS and before RNA extraction. It is not clear: what was the point of fixation, if RNA was isolated immediately after fixation? The point of fixation is to do it IN SITU, as soon as the muscle is taken out of the mouse, so that the subsequent enzymatic digestion of tissue does not cause activation. The fixation of tissue/cells is not mentioned in the legend of Fig. 3 and is not described in the RNA-seq section of the methods. This is important to correct. RNA extraction from fixed samples and obtaining high quality sequencing libraries is not trivial.

For how long were the tissue/cells fixed? What concentration of formaldehyde was used? Was formaldehyde quenched afterwards? How was the RNA extraction procedure adjusted to make it work with fixed samples? How were crosslinks reversed afterwards? The lack of clarity and the absence of all these details cast doubts on the rest of the RNA-seq data. The NCBI GEO entry GSE159026 for these samples also does not mention anything about fixation. In figure 3i, photos of representative stem cells are shown. They are circular, which according to Machado et al. is the shape of cells that were not fixed or fixed only after they were isolated.

We apologize for this ambiguity; no fixation was involved. We followed a conventional protocol of unfixed cell extraction followed by immediate RNA extraction. We have changed this in the manuscript on p10 lines 202/203 "To further address this at the phenotype onset, we analyzed freshly FACS-isolated *Nf1Myf5* and control p7 MPs by RNA-Seq."

The analyses leading to the heatmaps in Fig. 3E-H need to be much better described. What are the genes that compose these gene sets? They should be given as a supplementary table. What fraction of those is represented in the heatmaps as differentially expressed in mutant vs WT? How many more genes are NOT shown because not significantly different in the NF1 mutant? These results should also be shown as GSEA plots (with running enrichment score, etc), not only heatmaps. Also on these heatmaps the color legends are very hard to interpret because they are not symmetrical (e.g. the first one goes from -1.88 to 10.8, another goes from -5 to 10) and the choice of a three-colors palette makes little sense to show RPKM (estimated expression levels) values. The figure legend and the methods offer no help to guide the reader in properly interpreting these plots. Instead of showing RPKM, the authors should show log(fold change) or Z-score or median-centered RPKM values, which is customary with this sort of analyses. In this case, a three-colors heatmap color palette will make sense (below zero, zero, above zero). If they want, they can show also the average RPKM values as continuous variable row annotation.

We apologize for this shortcoming. We have now better explained how the custom gene sets have been generated and analyzed (page 11 lines 2018-226) and provide the gene lists as Excel tables (Supplementary Data 2). We furthermore provide information, how many of these genes are up- or downregulated, respectively in the above-mentioned paragraph. We clarified that the heatmaps only show a selection of these genes (out of the quiescence list, altogether 79 genes were upregulated, which would result in a very large heatmap). However, if the reviewer prefers, we could also show heatmaps for all significantly regulated genes (up- and down).

GSEA plots are shown for MyoD targets (Fig. 3c), we have now also shown additional GSEA analyses related to myogenic differentiation (Supplementary Fig. 3c). We also showed GSEA analysis for several categories related to ECM interaction, which comprise ECM genes associated with MuSC quiescence (Supplementary Fig. 3d), many of which shown in the heatmaps.

We have changed all heatmaps to Z-score.

The authors compare their dataset to that of Ryall et al. (ref 38). They should explain briefly what is that dataset and why it was chosen over the others presented in Fig. 3. Also they should explain in more details how the comparison (in Fig. S3E) was made. GSEA, using custom gene sets created from the Ryall et al data, could also be performed here.

We have attempted to improve the explanation in the text p13 lines 252-256” We finally compared our RNA-Seq data to the dataset from Ryall et al. 38 , comprising 2 month old MuSCs freshly isolated comparable to our protocol, and MuSCs that were kept for 2 days in culture to reflect activated cells. Comparison of normalized read counts confirmed a shift of Nf1Myf5 MPs transcriptome toward the signature of quiescent MuSCs (Supplementary Fig. 3f).”

In Fig. 4, the authors point that "condensed chromosome" was a significant gene set in their GSEA and thus conclude that this indicates chromatin alterations in the Nf1 mutants. This is over-simplistic. First, this is not indicative but "suggestive". Second, "condensed chromosome" contains 303 mouse genes (it is quite vast), including many falling under children terms associated with mitosis (e.g. the cohesin complex). So enrichment of "condensed chromosome" genes could simply be a reflection of a defective cell cycle: the mutants cycle less and thus express less genes associated with the M phase of the cell cycle.

Thank you for pointing this out; we have removed this.

Fig. 4B.: There are a few problems with this figure. It clearly shows two curves that do not overlap at the center, yet the authors wrote "...showed apparently unchanged H3K4me3 levels between controls and Nf1Myf5 MPs". Perhaps they were misled by the figure showing the average signal over an overtly broad 20 kb surrounding gene TSS. If they had chosen to focus on the TSS region (which is where H3K4me3 concentrates), by showing only ~1 kb on each side of it, the difference would be more obvious. Secondly, this is presumably the average signal for all genes in the genome (the authors forgot to give any detail anywhere). So there are possibly some genes showing higher signal and others

showing lower signal. Showing not only the mean but also the standard deviation would be very informative.

Thank you for pointing this out; we have replaced the panel with a depiction of 1kb surrounding the TSS in Fig. 4b (now Fig. 4a).

We have clarified the description in the text p14 lines 272-276 *“Chromatin immunoprecipitation sequencing (ChIP-Seq) analysis of freshly isolated p7 MPs suggested slightly decreased H3K4me3 levels around the transcriptional start site (TSS) averaged across all genes between controls and Nf1Myf5 MPs (Fig. 4a), and a global reduction of H3K27me3 levels in Nf1Myf5 MPs (Fig. 4b), which was confirmed by immunolabeling (Fig. 4c).”*.

The plots shown in Fig. 4a, b show averaged normalized coverage across all genes, and as such are descriptive. At least to our understanding it is not customary in the literature to use statistics on these depictions. We note that, based on a comment by reviewer 1, we have now performed a more in-depth analysis of H3K27me3 ChIP-Seq data. We used DiffBind to call differentially methylated regions and associated genes, and performed a functional analysis of this gene set including intersection with the RNA-Seq dataset (New Fig 4d, e). We have attempted to be as careful as possible with assumptions based on ChIP-Seq data (see also answer below, comment to quantitative analysis); we note, however, that the global downregulation of H3K27me3 was independently confirmed by immunolabeling including statistical analysis.

The authors wrote "acquisition of a quiescent state in Nf1Myf5 MPs is concomitant with reduced repressive H3K27me3 modification at quiescence-related genes." There are two problems with this statement. First, it is based on an analysis of all genes on the genome (Fig. 4c) and only two specific genes implicated in quiescence (Fig. 4d and S4b), which do not justify the level of generalization the authors are conveying. Second, the statement implies that there are no equivalent changes at genes with the opposite function (i.e. promoting proliferation). To support their statement, the authors should show the plot in Fig. 4C but broken down to show separately the situation at cell cycle promoting genes, and at cell cycle arrest genes.

We have deleted this statement. We also point out that in response to a concern raised by reviewer 1 we have improved our analysis of H3K27me3 data.

The ChIP-seq analysis method also does not explain how a quantitative analysis of H3K4me3 and H3K27me3 was made, e.g. there is no mention of internal standard spike-in chromatin to ensure that the WT and mutant datasets could be directly compared in a reliable manner. No control loci with invariant levels of these marks are shown either. The authors only mention "normalized ChIP seq profiles" with no procedural details.

We have now explained the ChIP-Seq data analysis in more detail in the methods section. We did not use spike-in, which, however, is not generally a standard procedure in ChIP-Seq analysis and was not custom at the time the analysis was planned and performed. We nevertheless acknowledge that the opinion of this is changing, we have therefore mentioned this as a limitation (p29 lines 556-559 *“We, however, note that our analysis did not employ e.g. chromatin spike-in. While we confirmed global deregulation of H3K27me3 and H4K16ac with alternative methods, quantitative assumptions at*

individual loci for all marks we analyzed should be taken with caution.”). We point out that for the histone marks we investigated further we confirmed global deregulation by immunolabeling.

In Fig. S4, the RPKM values calculated from Ryall et al. (GSE64379) and from this study cannot be directly compared, there are too many different variables (for instances the library preparation kits were different, Ryall was single-end (Ryall) vs paired end (this study)). What may be compared is the fold-change from each study (quiescent vs activated, for Ryall et al.)

We have removed this panel

For MeDIP-seq analysis (Fig. 4G), the authors need to better explain how the results were obtained and how to interpret them (what are odds ratio and the ratios shown (100/64582)?).

We have extended the figure legend as follows: “Enrichment of hypo- and hypermethylated regions among differentially methylated regions (DMRs) in control and Nf1Myf5 p7 MPs for different regions of interest (ROIs). Bar height corresponds to the odds ratio for the ROIs over all regions. The bars are labeled with the odds, e.g. the ratio between number of differential regions over the total number of regions of that ROI.”

The authors conclude from their epigenomic profiling that "epigenetic alterations observed in Nf1Myf5 MPs are consistent with a shift toward quiescence". However, although the validation work is nicely done, their analysis was quite superficial. A more accurate sentence would be "epigenetic alterations in Nf1Myf5 MPs, at the genes we subjected to validation, are consistent with a shift toward quiescence". It would also be important to better explain the rationale for selecting the specific metabolism genes that went through this validation: what links them to quiescence exactly?

Thank you very much for the suggestion, we have taken over this sentence.

There was no selection for specific metabolic genes; we note that metabolic genes were affected by differential DNA methylation and show one example from glycolysis and one from TCA cycle or electron transport chain, respectively, to illustrate this. In essence, the downregulation of these genes is, in our view, in line with a quiescence shift, as the latter is associated with reduced metabolic activity. This issue is elaborated in the following section of our manuscript.

The legend to Figure 5 needs to be more clear on how genes make it to be represented on the various heatmaps: are they all genes in these GO categories? Are they all significant differentially expressed genes? Are they the "core enrichment genes" from the GSEA analysis?

These heatmaps represent the significantly regulated genes from the GO terms; we have clarified this in the figure legend.

On page 17, the authors wrote "As Nf1Acta1 mice show no muscle defect, this suggests that perturbed muscle fiber metabolism in Nf1Myf5 animals 30 can be traced back to defects in juvenile MP".

In reality, the characterization of the Nf1Acta1 animals (Fig 2A, B, C) was superficial, focusing only on gross animal aspect and myofiber sizes. These animals (or their myofibers) were not analyzed in terms of their metabolism, they were not subjected to RNA-seq analysis, etc, so the statement has to either be removed or be backed with additional data for a direct comparison of the two strains (with Acta1-Cre and Myf5-Cre).

We agree with the reviewer that the analysis of Nf1Acta1 animals we performed was incomplete to cover these statements. We have extended the analysis of Nf1Acta1 mice, and now show that

- Muscle CSA and fiber diameter are unaltered also in the leg muscles TA and EDL (Fig. 2b, c);
- Nf1Acta1 animals have no fiber type shifts in the TA or EDL muscles (Fig. 2d), and
- Nf1Acta1 mice show no alteration in the expression of a panel of genes representative for glycolysis, OXPHOS and fatty acid metabolism (Supplementary Fig. 2e); all these genes we found deregulated in Nf1Myf5 muscle before (Wei et al. 2020 JCSM, doi: 10.1002/jcsm.12632).

We are confident that with these new data our statement is justified. We have changed the sentence as follows: *"As Nf1Acta1 mice showed no significant defect in muscle size, fiber types and expression of a panel of metabolic genes, this suggests that perturbed muscle fiber metabolism in Nf1Myf5 animals can be traced back to defects in juvenile MPs..."*.

In addition, to ameliorate our statement, we have changed the corresponding sentence in the discussion p32 lines 638/639 "Thus, in a muscle-specific NF1 mouse model, the muscular metabolic phenotype can be fully traced back to Nf1 activity in juvenile MPs.", we deleted the word "fully".

In the following sentence "This evidences transmission of differential gene expression signatures across cell divisions and cellular differentiation,..." we have replaced "evidenced" by "suggests".

We have also added the notion that Nf1Acta1 mice show no *obvious* muscle phenotype (p32 line 633).

Fig. 6J: the authors tested the effect of blocking the NOS pathway in Jagged-treated cells. However, they omitted to also treat similarly the No-Jagged control cells. Their conclusion that "pan-NOS inhibitor L-NAME cancelled the hyper-responsiveness to Jagged-1" is thus not fully supported, since in principle L-NAME could also act independently of Jagged-1. The statement should be revised, or new data showing that L-NAME has no effect in the absence of Jagged-1 stimulation should be presented.

Thank you very much for pointing this out, however, we only partially understand the comment. We specifically tested the response of the Notch pathway in MPs in this assay. Jagged1 treatment activates this response, thus we reasoned this to be the best situation to test potential inhibitors of this pathway. We nevertheless agree that a treatment of MPs with the inhibitors w/o Jagged stimulation may have been a useful control experiment.

We have rephrased the text: p24 lines 476-487 *"... pan-NOS inhibitor L-NAME cancelled the hyper-responsiveness to Jagged-1 (Fig. 6j), although we cannot formally exclude an effect of the inhibitors independent of Jagged1 treatment."*

page 23: The following statement is not fully supported by the data: "This indicated that aberrant Notch pathway activity was not abrogated but effectively blunted by our DAPT regime in Nf1Myf5 mice." Making the call between abrogation and blunting (taken to mean "partial decrease") would require more evidence than just qRT-PCR (e.g. showing that cells have a decreased response to Jagged-1). For one, it could be that the effect at the level of protein abundance is much more pronounced than at the steady state mRNA level.

Thank you for pointing this out, we agree. We have removed the sentence.

To make a link between the altered metabolism (reduced glycolysis) and the Notch pathway, the authors invoke the overlap between Sirt1 target genes (identified by Ryall et al.) and profiles of H4K16 acetylation (a mark erased by Sirt1). They conclude in the discussion "H4K16ac decoration is highly reduced in Nf1Myf5 juvenile MPs. Myogenic genes affected in Nf1Myf5 MPs overlap genes identified as Sirt1 targets in MuSCs, indicating Sirt1-mediated metabolic reprogramming of Nf1Myf5 juvenile MPs, preventing myogenic differentiation." However, the verb "indicate" feels too strong, considering the evidence presented: the authors do not offer any convincing validation of this hypothesis that Sirt1 is really involved in metabolic reprogramming in the Nf1Myf5 mice. Supporting evidence (more than correlative observations) would need to be provided, or the statement should be revised.

Thank you for pointing this out, we agree. We have changed the sentence **p31 lines 624-627** "H4K16ac decoration was reduced in Nf1Myf5 juvenile MPs. Myogenic genes affected in Nf1Myf5 MPs overlap genes identified as Sirt1 targets in MuSCs 38, thus it is possible that Sirt1-mediated metabolic reprogramming of Nf1Myf5 juvenile MPs may contribute to impairing their myogenic differentiation."

Other notes:

page 3: "and their transmission to quiescence." Probably the authors mean "transition"
Thank you, corrected.

page 5: "Proliferation rates of Nf1Myf5 Pax7+ cells assessed by Ki67 immunolabeling". Staining with Ki67 does not give the "rate of proliferation" but rather the proportion of cells actively proliferating. Growth curves would be needed to assess the proliferation rate.

Changed to "Proportions of proliferating of Pax7+ cells assessed by Ki67 immunolabeling of tissue sections showed a slight reduction in Nf1Myf5 muscle at p7,..."

Fig 1c: to be informative, the IF image should show the two color channels separately, as in Fig. 1B. The Pax7 intensity measurement should be a separate figure panel (with its own letter label). The legend does not explain what the horizontal black bars represent (the mean? the median?). Those data should be shown as a boxplot (or better, as a violin plot) to better represent the distribution of values.
We have changed the figure accordingly.

Fig. 1E: the y-axis label indicates "% myod+ nuclei in MF20+ cells", and shows a range from 0 to 0.6. These are very low percentages. Did the authors forget to multiply by 100? Or perhaps they meant "PROPORTION of myod+ nuclei...?"

Thank you for pointing this out! 0.6 should be 60%, we changed this.

Acta1-Cre: To avoid any misunderstanding, the authors should specify in the relevant methods section that this Cre driver is a transgenic mouse line expressing Cre recombinase under the control of the human alpha-skeletal actin promoter. Maybe the Jax stock number can be given too.

We have added the following on p8 lines 112-163 "...we inactivated Nf1 using Acta1Cre, which targets myofibers but not myoblasts³⁵ via expression of Cre from a transgene driven by the human skeletal actin promoter."

Fig 2d: The authors wrote that Nf1 is "barely detectable" in muscle lysates, but this feels like an overstatement. The protein is perfectly detectable and simply less expressed compared to myoblasts. We have changed this to "expressed at low levels".

The main text and methods should mention how many biological replicates were used for the RNA-seq experiment (supp figure 3 indicates 2 of each genotype).

We have added this to the main text as well as to the methods section (p10 lines 203/204, p38 line 815).

Figure 3: the panel letters in the figure and in the legend do not all match (e.g. panel d is GSEA in the figure but fluorescence signal in the legend).

Thank you, corrected.

Fig 3.A: the volcano plot show the $-\log_{10}$ of pvalue, but it is the pvalue after multiple hypothesis testing that must be shown instead.

We apologize for mislabeling; the p-values we used in this figure do indeed represent the Benjamini-Hochberg corrected p-value. This is corrected in the figure and stated in the figure legend.

Fig. 4G: the x-axis indicates "CpG insland", I presume it should be "island"

Thank you for spotting; corrected.

page 17: "and decreased H4K14ac decoration", it should be H4K16ac

corrected

The legend to Fig 6A should make it clear that the cells being analyzed are (e.g. "RNA extracted from MPs immediately FACS"). Otherwise the reader may get confused with panel 6B (MPs put in culture for 48h)

Thank you, changed to "RT-qPCR of Notch pathway component and target genes in RNA extracted from freshly FACS-isolated..."

Fig 6C: the multi-channel IF needs to be split into individual channels, because the green color (Pax7 signal) is strong strong, it hides all other colors.

We have added single channel images.

page 21: the authors omitted to explain what DAPT is, and what is the rationale for using it in their experiment.

We have added the following p26 lines 510-514: "To test whether Notch signaling hyperactivation in Nf1Myf5 MPs is needed for premature quiescence induction and long-term metabolic reprogramming in vivo, we treated Nf1Myf5 pups with 5 doses of 30 mg/kg of the Notch pathway inhibitor DAPT (or placebo control) from p6 to p18 (Fig. 7b). DAPT is an inhibitor of γ -Secretase, preventing Notch cleavage and thus signal transduction to the nucleus."

page 23: "myonucelar accrual"

We do not quite understand the comment; we have changed this to myonuclear accretion, is this what the reviewer suggested?

Reviewer #3 (Remarks to the Author):

The investigators use conditional Nf1 knockout mice to demonstrate that Nf1 is not required in muscle fibers, but rather postnatal myogenic progenitors. They further demonstrate that this requirement operates at the level of cell cycle exit, quiescence, and differentiation, which is dependent on Mek/Erk/Nos signaling. This Mek-driven effect sensitizes Nf1-deficient MPs to Notch signaling. Taken together, these observations extend what we know about neurofibromin regulation of metabolism.

Since patients with NF1 are heterozygous for a loss of function NF1 allele, it would be important to analyze MPs heterozygous for a loss of function Nf1 allele in order to be relevant to the human condition.

Thank you very much for the comment, we agree that this is an important point. Indeed, NF1 is an autosomal-dominant disorder. Although somatic second hits leading to loss of heterozygosity have been seen in Nf1-dependent tumors (see e.g. Brems et al. *Lancet Oncol* 2009), this is unlikely the case in musculoskeletal manifestations. However, mice with homozygous deletion of Nf1 are overall healthy (Brannan et al. *Genes Dev* 1994), yet they reflect some of the skeletal NF1 features as delayed fracture healing (Schindeler et al. *J Orthop Res* 2008), while other skeletal features are not reflected (see e.g. Sullivan et al. *Hum Mol Genet* 2013 and references therein). However, Nf1 heterozygous mice do not display any recognizable muscle phenotype (Sullivan et al. *Hum Mol Genet* 2013), thus in this case, Nf1^{+/-} mice do not model the human condition. This is not the first example of differences in humans versus mouse models with respect to penetrance and expressivity of genetic traits. We have ourselves actually seen the same before for hand malformation syndromes (Schwarzer et al. *Hum Mol Genet* 2008, Witte et al. *PNAS* 2010).

It thus appears that, in humans, myogenic cells are more susceptible to alterations in Nf1-dependent signaling than murine myogenic cells. This discrepancy made it necessary to create a homozygous mouse model, which, due to the early lethality of Nf1^{-/-} embryos, needed conditional inactivation.

Experimentally, we have addressed the issue as follows:

- we have isolated MPs from control (Myf5Cre-negative, Nf1^{flox/+}), and Nf1 haploinsufficient (Myf5Cre⁺, Nf1^{flox/+}) animals and performed RT-qPCR for *Nf1* itself (Supplementary Fig. 1b), which did not show a change in expression between wild type and haploinsufficient mice (as we have previously observed in Nf1Myf5 whole muscle tissue). This suggests that in mice compensatory mechanisms upregulate Nf1 expression from the remaining allele to normal levels, thus likely precluding a haploinsufficiency phenotype.
- performed RT-qPCR for quiescence genes *Pax7*, *Calcr* and activation genes *Myog*, *Myh3*, which did not show alterations in haploinsufficient MPs either (Supplementary Fig. 3g)
- performed RT-qPCR for glycolytic genes *Hk2* and *Pfkfb1* (Supplementary Fig. 5a) and Notch pathway components *Hes1*, *Hey1* and *Notch1* (Supplementary Fig. 7d), demonstrating unaffected expression.
- Furthermore, we performed cytospin analysis of control and Nf1 haploinsufficient MPs followed by immunolabeling for Pax7 and Ki67, and Pax7 and MyoD; no alterations in the proliferation rate or the Pax7/MyoD ratio was observed (Supplementary Fig. 1d).

We hope that with this analysis we could adequately address the reviewer's concern.

1. Given the reduction in Myf5 levels in these mice, were Myf5-Cre mice identical to wild-type mice in terms of MP gene expression and function?

This has been addressed in our previous paper (Wei et al. 2020 *JCSM*, doi: 10.1002/jcsm.12632; p1764: "Note that, although the Myf5Cre allele is a loss of function, *Myf5* mRNA expression in whole muscle tissue both during development (E18.5) and at p21 was indistinguishable between Myf5^{+/+} and Myf5Cre^{+/+} mice (Supporting Information, Figure S2C), indicating compensation by the intact allele.").

In the present manuscript, unaltered expression of *Myf5* in wild type and *Myf5Cre/+* p7 MPs is shown in Supplementary Fig. 1c. Furthermore we point out that we used *Myf5Cre;Nf1flox/+* mice as controls, thus the *Myf5* allele is excluded as confounder in the comparative analysis. We have made this clearer in the manuscript on p5 lines 104-106: “*Nf1* expression was unaltered in MPs of *Nf1*-haploinsufficient *Myf5Cre;Nf1flox/+* mice (Supplementary Fig. 1b), which were used as controls for all further experiments.”.

2. Figure 1. Was there any change in MP apoptosis or premature senescence?

No alterations in apoptosis is shown in Supplementary Fig. 1e. We did not directly address cell senescence, however this never showed up as deregulated category in the analysis of RNA-Seq data. We also note that *Cdkn2a*, encoding p16^{INK4A}, widely used as senescence marker, is expressed at low levels (RPKM below 1) in control or *Nf1*-deficient p7 MPs (see Supplementary Data 1).

3. Figure 2. Size markers should be included for all Western blots. What is the molecular size of the neurofibromin band shown?

We have now indicated band sizes

4. Figure 4. Is N-Ras the predominant hyperactivated Ras in *Nf1*-deficient MPs? Some of the changes in RNA expression, while statistically different, look minimally different (<10-20% relative to controls).

We show this GSEA plot to illustrate increased RAS signaling, not to pinpoint one specific branch of this pathway. We have rephrased this sentence to make this point clearer on p10 lines 205/206: “Gene set enrichment analysis (GSEA) showed an enrichment for NRAS Signaling in *Nf1Myf5* MPs (Fig. 3b) in line with upregulated RAS pathway activity”.

Indeed, *Nras* and *Hras* were the predominantly expressed Ras genes (RPKM around 20) in p7 MPs; *Kras* was not detected in our RNA-Seq.

We are not sure, which minimally different RNA expression levels the reviewer is referring to; in the global RNA-Seq analysis, only genes with log2 fold change above 2 and below 0,5 were considered; in Fig. 4f, the lowest log2 fold change (*Tet1*) is 1, meaning a 100% upregulation.

5. The results in panel j should be normalized to total p70S6K, rather than actin, to determine the relative reduction in activated p70S6K in *Nf1*-deficient MPs.

We would like to point out that performing new western blots in triplicates would require the production of 3 *Nf1Myf5* mutants and 3 controls; on average we have one mutant per litter, thus approx. 18-24 offspring animals would be produced solely for this analysis. In the light of 3R principles we would like to avoid this. We think that controlling for equal protein loading via actin allows to quantitatively assess absolute phosphor-p70S6K levels. Nevertheless, we agree that using total p70S6K would be a better control. We point out that we also confirmed increased TORC1 activity by immunofluorescence analysis for phosphor-S6 (Fig. 3k). Since this point of data is not integral for our manuscript we could remove this panel if the reviewer insists.

6. Since neurofibromin has previously been shown to result in increased mTOR signaling, which should increase p70S6K phosphorylation, what do the authors think is responsible for this finding?

Indeed, mTORC1 has been seen activated in other Nf1 deficient cell systems. We nevertheless found reduced mTORC1 signaling in Nf1-deficient p7 MPs (this study) and muscle tissue (Wei et al. 2020 JCSM, doi: 10.1002/jcsm.12632). At present we do not have an explanation for this, the simplest explanation would be an energy deficit upregulating AMPK activity, which we in fact observed in Nf1-deficient muscle tissue (discussed in Wei et al. 2020 JCSM, doi: 10.1002/jcsm.12632). We, however, do not know whether AMPK activity is deregulated in Nf1-deficient p7 MPs. We expect, based on the general low energy demand of quiescent MuSCs, this may not be the case. Another possible explanation can be found in our new in silico metabolic flux modeling. These data suggest a decreased capacity of Nf1-deficient MPs for uptake of branched chain amino acids including leucine, which is a major positive regulator of mTORC1 activity. This, however, is pure speculation at this point, which we would rather not include in the manuscript.

REVIEWER COMMENTS

Reviewer #2 (Remarks to the Author):

The authors have addressed all my comments in a satisfactory fashion and I feel that their manuscript has significantly improved as a result.

Reviewer #3 (Remarks to the Author):

The revised manuscript addresses many of the concerns and critiques raised by the reviewers, including attention to detail, inadvertent errors, and misstatements. However, several concerns remain.

First, it is not true that “mice with homozygous deletion of Nf1 are overall healthy”. In fact, these mice die before E12.5 of a cardiac vessel malformation and no live Nf1^{-/-} mice reach birth.

Second, homozygous conditional KO mice may indeed not accurately model the human condition. It would be critical to determine whether loss of heterozygosity exists in human NF1-related myopathy, as has been reported for many other NF1 clinical features, such as café-au-lait macules and tibial pseudarthrosis. Given their findings and conclusions regarding “compensatory mechanisms”, it is still not clear what relevance this model has to the human condition.

Third, the Western blots remain problematic to interpret without appropriate loading controls. The inclusion of additional non-normalized data (e.g., immunofluorescence) does not add much to bolster the conclusions.

Reviewer #4 (Remarks to the Author):

The authors present their revisions for the manuscript entitled, ‘Neurofibromin 1 controls metabolic balance and Notch-dependent quiescence of juvenile myogenic progenitors’.

The authors have carried out substantial re-organization of the manuscript as well as new data analysis to address the comments of Reviewer #1. Many of the concerns relating to lack of specificity and detail in ChIP-seq experiments were also raised by reviewer #2, highlighting their importance. However, some further work s required to fully satisfy the reviewer's concerns (my comments indicated as 'Extra Reviewer'):

Figure 2D, in the western blot of whole muscle, it is surprising that the mutant (myf5-cre; nf1-flox) and control muscle only show subtle difference of nf1 protein expression. The authors provided a quantification in Figure 2E. It is still difficult to reconcile this modest difference in the protein levels to the major myofiber phenotype shift in the previous report. Maybe the residue nf1 protein of the mutant mice is from other cells in the muscle tissue, i.e. fibroblast? Or it is from the non-myf5 lineage muscle cells?

Thank you very much for this comment. Indeed, in Fig. 2d (now supplementary Fig. 2g) shows residual Nf1 protein expression in Nf1Myf5 homozygous KO muscle, thus suggesting only a moderate downregulation. We have

- analyzed Nf1 mRNA expression during myogenic differentiation of primary myoblasts (Fig. 2d) reconfirming the downregulation of Nf1.

- We have also analyzed Nf1 mRNA levels in whole muscle tissue versus isolated primary myoblasts and fibroblasts (Fig. 2e), showing exactly what the reviewer suggested; Nf1 is mainly expressed in fibroblasts in muscle tissue, thus explaining the residual expression in Nf1Myf5 mice, where fibroblasts are unaffected.

Extra Reviewer: "The original western blot (now supplementary Fig. 2g) must be returned to the main figure for comparison with the qPCR results (Fig. 2e), and the moderate effect of knockdown clearly noted in the corresponding results section. The authors above refer to Figure 2e, but do they mean Fig 2f?"

The H3K27ac enrichment in quiescent muscle stem cells is still controversial. Some studies show that the H3K27me3 marked heterochromatin is the hallmark of the quiescence in muscle stem cells. In this study, the global H3K27me3 was dramatically reduced around TSS in nf1 KO cells (Fig 4C), but the authors only picked up two genes (pax7 and cdkn2a) to suggest the reduced H3K27me3 is associated with the

impairment of proliferation. They should perform thorough functional annotation analysis of all the H3K27me3 target genes and whether H3K27me3 depletion on the TSS can increased the overall expression of the target genes.

Thank you very much for pointing this out, we apologize for the shortcoming. We have now improved this analysis by -performing DiffBind analysis to identify genes with significantly reduced H3K27me3 in Nf1Myf5 MPs; we provide this gene list as Supplementary Data 3, and performed GO analysis (Fig. 4d). We have furthermore intersected these genes with genes upregulated in Nf1Myf5 MPs (Fig. 4e; gene list in Supplementary Data 4) and performed GO analysis on the overlapping genes (Supplementary Fig. 4a). This comment was particularly helpful: Pax7, which we picked to illustrate reduced apparent H3K27me3 was indeed contained in this intersected gene list (Supplementary Data 4; Fig. 4e).

Since the overlap between H3K27me3 genes and upregulated genes was, however, low, we added the following sentence (p14 lines 279-282): "Intersecting genes with reduced H3K27me3 levels with genes upregulated in Nf1Myf5 p7 MPs showed only a low overlap of 248 genes (Fig. 4e; gene list in Supplementary data 4), suggesting that reduced H3K27me3 alone cannot explain gene deregulation in Nf1Myf5 MPs."

Extra Reviewer: "The new analysis is useful. The authors should clearly show the base line differences in Fig 4a and Fig 4b (i.e. The results of their IgG/input controls) for comparison."

Extra Reviewer: "In figure 4K, how was the boxed region chosen? Why is it important, appear a long way from the promoter/TSS which is defined as +-500 TSS. Generally, th authors need to improve their definitions when referring to promoters/ CpG Islands, etc. How were they defined, and why was that exact definition chosen?"

Extra Reviewer: "Figure 4i needs more explanation, what do the ratios mean. I find it hard to understand how so many CpG Islands are enriched for hypomethylation/hypermethylation but not promoters given that 70% of promoters contain CpG Islands."

Extra Reviewer: "The authors should not use the term hypermethylation or hypomethylation as meDIP gives no indication of absolute methylation values. Increased or decreased enrichment should be used."

The authors claimed the TSS H3K4me3 distributions are similar between KO and WT cells, but H3K27ac are very different, please provide statistical and significance analysis.

The plots shown in Fig. 4a, b show averaged normalized coverage across all genes, and as such are

descriptive. At least to our understanding it is not customary in the literature to use statistics on these depictions. We note that, as elaborated above, we have now performed a more in-depth analysis of H3K27me3 CHIP-Seq data. We have attempted to be as careful as possible with assumptions based on CHIP-Seq data; we note, however, that the global downregulation of H3K27me3 was independently confirmed by immunolabeling including statistical analysis. We have further clearly noted the limited quantitative information of our CHIP-Seq data in the discussion (p29 lines 556-559 “We, however, note that our analysis did not employ e.g. chromatin spike-in. While we confirmed global deregulation of H3K27me3 and H4K16ac with alternative methods, quantitative assumptions at individual loci for all marks we analyzed should be taken with caution.”).

Extra Reviewer: "This statement should be moved to the relevant results section"

Response to reviewer comments

Reviewer #2 (Remarks to the Author):

The authors have addressed all my comments in a satisfactory fashion and I feel that their manuscript has significantly improved as a result.

Thank you very much for the positive appreciation of our revised manuscript.

Reviewer #3 (Remarks to the Author):

The revised manuscript addresses many of the concerns and critiques raised by the reviewers, including attention to detail, inadvertent errors, and misstatements. However, several concerns remain.

First, it is not true that “mice with homozygous deletion of Nf1 are overall healthy”. In fact, these mice die before E12.5 of a cardiac vessel malformation and no live Nf1^{-/-} mice reach birth.

We seriously apologize for this mistake in the reply to the reviewer’s questions, we intended to write “mice with *heterozygous* deletion of Nf1 are overall healthy”. We note that in the manuscript, this had been stated correctly: P4 lines 78/79 read “Constitutive inactivation of Nf1 causes early embryonic lethality, while Nf1 haploinsufficiency does not affect muscle development or function in mice”

Second, homozygous conditional KO mice may indeed not accurately model the human condition. It would be critical to determine whether loss of heterozygosity exists in human NF1-related myopathy, as has been reported for many other NF1 clinical features, such as café-au-lait macules and tibial pseudarthrosis. Given their findings and conclusions regarding “compensatory mechanisms”, it is still not clear what relevance this model has to the human condition.

We agree with the reviewer, that a homozygous loss of function model may not accurately model human NF1. We do not have access to human samples at this point; it was reported (Summers et al. 2018, DOI: 10.1093/hmg/ddx423) that NF1 protein expression in human muscle samples was reduced roughly by 50% suggestive of haploinsufficiency. Nf1 heterozygous mice, however, do not show muscle defects (Sullivan et al. 2014, DOI: 10.1093/hmg/ddt515). Thus, in muscle, there is a clear species difference, making the appreciation of an animal model inherently difficult, and requiring homozygous inactivation in the mouse.

We stress that the primary aim of this study was not to exactly model the human condition, but to elucidate Nf1 function in myogenic cells. Neurofibromatosis type 1 is characterized by the systemic loss of one NF1 allele, and even at sites of LOH, a bystander effect for surrounding haploinsufficient cells has been shown (see e.g. in Frost et al. DOI: 10.20517/jtgg.2022.14). Thus, no animal model targeting Nf1 in a specific cell type will accurately model human NF1, yet this approach is necessary to disentangle the function of Nf1 in this specific cell type.

Nevertheless, we agree that this is a limitation of the study. We have therefore added the following paragraph to the manuscript (p33 lines 632-644): “NF1 is an autosomal-dominant disorder featuring systemic Nf1 haploinsufficiency; however certain features such as neurofibromas or pseudarthroses can harbor loss-of-heterozygosity (LOH) ⁸¹. Analysis of NF1 protein in human NF1 muscle samples showed an approximate 50% reduction in line with haploinsufficiency ³², however LOH in only specific cell type(s) of muscle tissue cannot be excluded. Neither muscle-haploinsufficient Myf5Cre;Nf1flox/+ mice (this study) nor Nf1^{+/-} mice ²⁹ show a discernable muscle phenotype. Overall, Nf1 haploinsufficient mice only in part recapitulate the spectrum of human NF1 haploinsufficiency-related features ⁸¹, requesting homozygous inactivation of Nf1 to model disease manifestations, even if in

humans these are not associated with LOH^{29,82}. Indeed, our data suggest a compensatory upregulation of Nf1 expression from the intact allele in MPs (this study), and muscle tissue³¹ in mice. Nevertheless, while a homozygous Nf1 inactivation in mouse model may not recapitulate the exact genetic background seen in NF1 patients, this model allows to dissect the function of Nf1 in a specific tissue type, in this case muscle and myogenic progenitors.”

Third, the Western blots remain problematic to interpret without appropriate loading controls. The inclusion of additional non-normalized data (e.g., immunofluorescence) does not add much to bolster the conclusions.

We reckon this refers to the blot shown in Fig. 3j, which had been criticized by the reviewer before. Instead of unphosphorylated p70S6K we had used actin as loading control. We follow the reviewer’s recommendation and have replaced this blot by one using unphosphorylated p70S6K as control.

Reviewer #4 (Remarks to the Author):

The authors present their revisions for the manuscript entitled, ‘Neurofibromin 1 controls metabolic balance and Notch-dependent quiescence of juvenile myogenic progenitors’.

The authors have carried out substantial re-organization of the manuscript as well as new data analysis to address the comments of Reviewer #1. Many of the concerns relating to lack of specificity and detail in ChIP-seq experiments were also raised by reviewer #2, highlighting their importance. However, some further work s required to fully satisfy the reviewer’s concerns (my comments indicated as 'Extra Reviewer'):

Figure 2D, in the western blot of whole muscle, it is surprising that the mutant (myf5-cre; nf1-flox) and control muscle only show subtle difference of nf1 protein expression. The authors provided a quantification in Figure 2E. It is still difficult to reconcile this modest difference in the protein levels to the major myofiber phenotype shift in the previous report. Maybe the residue nf1 protein of the mutant mice is from other cells in the muscle tissue, i.e. fibroblast? Or it is from the non-myf5 lineage muscle cells?

Thank you very much for this comment. Indeed, in Fig. 2d (now supplementary Fig. 2g) shows residual Nf1 protein expression in Nf1Myf5 homozygous KO muscle, thus suggesting only a moderate downregulation. We have

- analyzed Nf1 mRNA expression during myogenic differentiation of primary myoblasts (Fig. 2d) reconfirming the downregulation of Nf1.

- We have also analyzed Nf1 mRNA levels in whole muscle tissue versus isolated primary myoblasts and fibroblasts (Fig. 2e), showing exactly what the reviewer suggested; Nf1 is mainly expressed in fibroblasts in muscle tissue, thus explaining the residual expression in Nf1Myf5 mice, where fibroblasts are unaffected.

Extra Reviewer: "The original western blot (now supplementary Fig. 2g) must be returned to the main figure for comparison with the qPCR results (Fig. 2e), and the moderate effect of knockdown clearly noted in the corresponding results section. The authors above refer to Figure 2e, but do they mean Fig 2f?"

We thank the reviewer for this comment and agree that this would be a helpful depiction. However, we have now decided to remove the western blots in Fig. S2f and g. These blots do not meet the quality criteria of the journal, merely based on the fact that we did not use a fluorescent marker ladder that would be visible on the blot images. We note that the key messages from these two blots (decreasing

Nf1 expression upon myoblast differentiation in Fig. S2f, and low expression of Nf1 in myofibers compared to myoblasts, Fig. S2g) are still backed by RT-qPCR analysis (Fig. 2a, b).

Concerning the moderate effect of knockdown observed by western blot in muscle tissue of Nf1Acta1 mice: to explain this, we had previously checked Nf1 gene expression in different cell types of muscle tissue and show (Fig. 2f, now Fig. 2b) that in mature muscle the majority of Nf1 message is expressed in muscle-resident fibroblasts, which are not targeted by Acta1-Cre, thus explaining residual Nf1 protein in muscle of Nf1Acta1 mice. This, however, does not at all compromise our results, as the intention of this experiment was to delete Nf1 specifically in mature muscle fibers, not myoblasts or connective tissue fibroblasts.

Concerning the second remark, there appears to be a misunderstanding. Fig 2e (now Fig. 2a) shows the decrease of Nf1 gene expression, while Supplementary Fig. 2f (now removed, see above) showed decrease of Nf1 protein expression during myogenic differentiation. Supplementary Fig. 2g (now removed, see above) confirmed low expression of Nf1 in muscle tissue compared to myoblasts, but also showed residual expression in Nf1Acta1 mice, backed by RT-qPCR data shown in Fig. 2f (now Fig. 2b) demonstrating fibroblasts as major source of this expression. As noted, due to quality reasons we have now removed the protein expression data; we have also completely reorganized this figure and the corresponding text to increase clarity.

The H3K27ac enrichment in quiescent muscle stem cells is still controversial. Some studies show that the H3K27me3 marked heterochromatin is the hallmark of the quiescence in muscle stem cells. In this study, the global H3K27me3 was dramatically reduced around TSS in nf1 KO cells (Fig 4C), but the authors only picked up two genes (pax7 and cdkn2a) to suggest the reduced H3K27me3 is associated with the impairment of proliferation. They should perform thorough functional annotation analysis of all the H3K27me3 target genes and whether H3K27me3 depletion on the TSS can increased the overall expression of the target genes.

Thank you very much for pointing this out, we apologize for the shortcoming. We have now improved this analysis by -performing DiffBind analysis to identify genes with significantly reduced H3K27me3 in Nf1Myf5 MPs; we provide this gene list as Supplementary Data 3, and performed GO analysis (Fig. 4d). We have furthermore intersected these genes with genes upregulated in Nf1Myf5 MPs (Fig. 4e; gene list in Supplementary Data 4) and performed GO analysis on the overlapping genes (Supplementary Fig. 4a). This comment was particularly helpful: Pax7, which we picked to illustrate reduced apparent H3K27me3 was indeed contained in this intersected gene list (Supplementary Data 4; Fig. 4e).

Since the overlap between H3K27me3 genes and upregulated genes was, however, low, we added the following sentence (p14 lines 279-282): "Intersecting genes with reduced H3K27me3 levels with genes upregulated in Nf1Myf5 p7 MPs showed only a low overlap of 248 genes (Fig. 4e; gene list in Supplementary data 4), suggesting that reduced H3K27me3 alone cannot explain gene deregulation in Nf1Myf5 MPs."

Extra Reviewer: "The new analysis is useful. The authors should clearly show the base line differences in Fig 4a and Fig 4b (i.e. The results of their IgG/input controls) for comparison."

We acknowledge that the use of IgG/input controls in ChIP-seq experiments to account for non-specific background signals and to provide a point of comparison can be useful, especially for validating read coverage generated by non-verified antibodies. Since we used ChIP-verified histone modification antibodies that have been used by multiple labs before generating specific enrichment, for the experiments presented in Figures 4a and 4b, we unfortunately did not sequence input or IgG controls. We fully acknowledge this limitation and recognize that input controls would have provided a clearer baseline for comparison. We have extended the previous statement of limitation to also cover this point, which now reads (p14 lines 270-272): "We, however, note that our analysis did not employ IgG or input analysis, or e.g. chromatin spike-in, thus quantitative assumptions at individual loci should be

taken with caution.”. We also point out that the ChIP-Seq analyses we performed were not used for hypothesis generation or to implement a mechanism explaining transcriptional changes we observed, but as a correlative indication of a quiescence shift of Nf1Myf5 myogenic progenitors. We also note that the global downregulation of H3K27me3 was independently confirmed by immunofluorescence.

Extra Reviewer: "In figure 4K, how was the boxed region chosen? Why is it important, appear a long way from the promoter/TSS which is defined as +/-500 TSS. Generally, the authors need to improve their definitions when referring to promoters/ CpG Islands, etc. How were they defined, and why was that exact definition chosen?"

The box was inserted to facilitate comparison between control and Nf1Myf5 tracks concerning the peaks highlighted by arrows. We realize that our labeling with a box and the arrows was confusing and double, we have removed the arrows.

Indeed, the region pointed out is not within the immediate +/-500bp region surrounding the TSS, which we used as promoter annotation (also see answer below), nevertheless these peaks still fall into a relatively narrow upstream window of less than 5kb in both examples. We point out that we do not claim mechanistically that the differential methylation observed at this region is responsible for the downregulation of the respective genes and refrain from speculation.

We have attempted to define our criteria as clearly as possible; in the materials and methods section, we have defined “Differentially methylated regions (DMRs) where called at 250 base windows at an FDR of 10%, and annotated with promoter (TSS +/- 500 bases), exonic, and gene body regions as well as model based CpG islands (CGI) [102]”. We have added the definition used for promoters to the respective figure legend for clarity. We note that promoter size cannot be universally defined and is inherently a compromise; given that promoters are typically annotated anywhere between 100 bp and 1 kb, we chose 500 bp as such. We point out that the model used for CpG island calling is the same model used by the USCS genome browser (Ref. 102), so we are confident we follow recognized standards in defining these regions.

Extra Reviewer: "Figure 4i needs more explanation, what do the ratios mean. I find it hard to understand how so many CpG Islands are enriched for hypomethylation/hypermethylation but not promoters given that 70% of promoters contain CpG Islands."

Thank you for the comment, indeed, this is somewhat unexpected; nevertheless, this could be due to the fact that we used +/- 500 bp regions defining promoters, thus, apparently, DMR CpG islands were preferentially located not in direct vicinity of the transcriptional start site. The figure depicts odds ratios, meaning we compare the ratios of numbers of DMRs (253) found in all regions analyzed (2231983) (odds = $235/2231931 = 0.011\%$) to odds in specific regions, e.g. promoter regions ($3/33919 = 0.009\%$), thus the odds ratio of DMRs in promoters vs all regions is $0,009/0,011=0,818$. For 64582 CpG islands we find 100 DMRs, meaning a ration of 0,10; thus, compared to all regions this provides an odds ratio of $0,15/0,011=13,64$. Thank you for pointing out that our description was hard to follow, we have attempted to clarify this in the figure legend and the main text (p15, 18).

Extra Reviewer: "The authors should not use the term hypermethylation or hypomethylation as meDIP gives no indication of absolute methylation values. Increased or decreased enrichment should be used."

Thank you for your feedback regarding our use of the terms "hypermethylation" and "hypomethylation" in relation to MeDIP-seq data. We understand your perspective that these terms might be perceived as indicating absolute methylation values. While our intention was to convey "hypermethylation" as a relative increase in methylation levels compared to the wildtype, we recognize the potential for ambiguity. To avoid any confusion and in consideration of your suggestion,

we now use the terms "increased/decreased levels of methylation" or e.g. "gain of methylation" in our manuscript. We believe this adjustment will ensure clarity for our readers.

The authors claimed the TSS H3K4me3 distributions are similar between KO and WT cells, but H3K27ac are very different, please provide statistical and significance analysis.

The plots shown in Fig. 4a, b show averaged normalized coverage across all genes, and as such are descriptive. At least to our understanding it is not customary in the literature to use statistics on these depictions. We note that, as elaborated above, we have now performed a more in-depth analysis of H3K27me3 ChIP-Seq data. We have attempted to be as careful as possible with assumptions based on ChIP-Seq data; we note, however, that the global downregulation of H3K27me3 was independently confirmed by immunolabeling including statistical analysis. We have further clearly noted the limited quantitative information of our ChIP-Seq data in the discussion (p29 lines 556-559 "We, however, note that our analysis did not employ e.g. chromatin spike-in. While we confirmed global deregulation of H3K27me3 and H4K16ac with alternative methods, quantitative assumptions at individual loci for all marks we analyzed should be taken with caution.").

Extra Reviewer: "This statement should be moved to the relevant results section"

Thank you for the suggestion, we have taken this over and shifted this note to p14 lines 270-272.

REVIEWERS' COMMENTS

Reviewer #3 (Remarks to the Author):

The authors have addressed my concerns to the best of their abilities.

Reviewer #4 (Remarks to the Author):

The authors have largely addressed my concerns and I have no further comments.